# Influence of sand content on mechanical properties of sand-silt mixtures from check dam deposits in the Loess Hilly of Ningxia, China

Liucheng Chang☉, Ya Wang☉, Hongyu Wang🆔*

School of Civil and Hydraulic Engineering, Ningxia University, Yinchuan, Ningxia, P R of China

☉ These authors contributed equally to this work.
* why.nxts@163.com

**Data Availability Statement:** All relevant data are within the paper and its Supporting information files.

## Abstract

An accurate description of the stress-strain relationships of sand-fine mixtures is very important to analyze the soil's mechanical properties. Hence, a series of consolidated drained (CD) triaxial tests were performed on reconstructed sand-silt mixtures with sand contents of 0%, 16.67%, 28.57%, 50%, and 60% in the paper to examine the effect of the sand content on the stress-strain curves of the soil. Results show that for sand-fine mixtures with different sand contents, the stress-strain curves are also mainly strain softening though there exist different degrees of softening. In order to quantitatively describe the strain-softening characteristics of sand-fine mixtures, a modified Duncan-Chang model was developed. To verify the applicability of the modified mode, examples such as coral clay and undistorted loess are described and predicted. There is a high consistency between theoretical and experimental values. Finally, a sand-content-dependent constitutive model that considered the effects of sand content and confining pressure was proposed based on the modified Duncan-Chang model by constructing the relationship between model parameters and confining pressure and sand content. The constitutive model was implemented in ABAQUS software and verified by comparing the calculated results with the triaxial test data of sand-fine mixtures under the confining pressure of 500 kPa. The comparison results indicate that the constitutive model can reflect the real characteristics of sand-fine mixtures.

## Introduction

Currently, many check dams are in danger that can no longer meet the requirements of sand retaining and water storage because of low design standards, weak maintenance measures, and bulky sediments silting up the reservoir area in the early stage [1]. To ensure check dam function regularly, reinforced measurements are urgently needed. Damming on the reservoir alluviums in front of the dam is widely used in the reinforcement of dangerous check dams in Shaanxi, Ningxia, Gansu, and the rest of Loess Plateau since it can save filling material, shorten the construction period, and reduce product cost. The reservoir alluviums encountered in the

**Funding:** Thanks to the Natural Science Foundation of China (Project No.41962016) and the Doctoral Scientific Fund Project of the Ministry of Education of China (Project No.20136401110003). These funders play an important role in the study design, data collection and analysis, and preparation of the manuscript.

**Competing interests:** The authors have declared that no competing interests exist.

Loess Hilly of Ningxia are silt and clay with different sand contents [2–4] (larger than the 2mm sieve). The sand particles in the clay-silt-sand mixtures help reduce point-to-point contact between the fine particles (silt and clay) [5, 6], lowering the interactive force between them, which is one of the causes of dam failure. In order to analyze the stability of dams, the impact of sand content on the mechanical properties of clay-silt-sand mixtures must be investigated. Due to the particle size of clay and silt being less than 0.075mm, they can be classified as fine particles, and clay-silt-sand mixtures can also be referred to as sand-fine mixtures [7]. Several studies have been performed on the effect of sand content on the engineering characteristics of sand-fine mixtures. Payan et al. [8] conducted a comprehensive set of resonant column tests to investigate the influence of sand content on the small-strain Young's modulus of sand-fine mixtures and found that sand characteristics significantly affect the dynamic behavior of the soil at high sand contents and that over a certain percentage of sand content, the main factor controlling Young's modulus is the silt inclusion. Li and Tang [7] studied the influences of low fines content and fines mixing ratio on the undrained static shear strength of sand–silt–clay mixtures and concluded that sand particles' filling, bonding, and soil skeleton properties could all be influenced by fine particles simultaneously. Miftah et al. [9] investigated the effects of fine content on the undrained shear response of sand-clay mixtures. By performing a series of direct shear tests on sand-clay mixtures with fines contents varying from 0% to 25%, they found that the internal friction angle increased with increasing fines until it reached a maximum value of 45.68˚ at the transitional fines content (10%). Zhou et al. [10] demonstrated that sand content can have a significant effect on the liquefaction resistance of sand-silt mixtures. Ghadr et al. [11] carried out a series of undrained cyclic triaxial shear tests on fiber-reinforced silty sands and reported that the efficiency of fiber reinforcing decreases as the median size of the sand increases for sands with >40% silt content. Roshan et al. [12] examined the impact of fiber reinforcement on the durability of lignosulfonate-stabilized clayey sand under the wet-dry cycling condition and revealed that the addition of fibers increased the durability of stabilized clayey sand up to 12 cycles of wet-drying by 0.8%. Afrakoti et al. [13] evaluated the effect of coal wastes on the mechanical properties of cement-treated sandy soil. However, few experimental and theoretical works have been done on the effect of sand content on the machine of sand-fine mixtures.

The stress-strain relationships of geomaterials are the reflection of their mechanical responses to loads. The constitutive model that was chosen for the calculation has a significant impact on how reliable the geotechnical numerical results are. Consequently, the primary purpose of constitutive models is to provide an exact description of the soil stress-strain behavior, which is crucial for both the economics and safety of check dams built on the reservoir alluviums in front of the dam. In geotechnical engineering, there are numerous sophisticated soil models that are formed based on elastoplastic theory [14]. These models are theoretically rigorous, but they frequently ask for a number of expensive and challenging-to-quantify parameters [15]. Therefore, simpler models, such as the Duncan-Chang model proposed by Duncan and Chang, are preferable in some situations, especially when the stress path is not unduly convoluted or when there is a lack of sufficient data on the soil properties to conduct an exploratory investigation [15]. The Duncan-Chang model, which is developed based on the hypothesis that the stress-strain relations of soils follow the hyperbolic function, is a typical nonlinear elastic model [16]. The model has been widely utilized in the computation and simulation of stress-strain relations for various soils because of its benefits of few parameters, unambiguous physical meaning, and simple structure [17]. This model can adequately fit the weak hardening stress-strain curve. However, it has significant measurement errors when describing the strong hardening curve and is unable to represent softening curves [14]. The stress-strain curves of some geo-materials, such as Lake Agassiz clay [18], cemented paste

backfill [19], and cement-treated zinc-contaminated clay [20] are also mainly strain softening, although they show different degrees of softening. The stress-strain relation curves of sand-fine mixtures with different sand contents under various confining pressures exhibit the same rules in the triaxial test. In order to apply the Duncan-Chang model to the quantitative analysis of the effect of sand content on the mechanical properties of sand-fine mixtures, the model must be further modified.

In this paper, a modified Duncan-Chang model was proposed based on the original Duncan-Chang model for characterizing the softening stress-strain curves of geomaterials. On the basis of the modified Duncan-Chang model and the triaxial test results of sand-fine mixtures under the confining pressures of 100, 200, 300, and 400 kPa, a sand-content dependent constitutive model, considering the effect of the sand content and confining pressure on the mechanical characteristics of the soil by constructing the relationship between model parameters and confining pressure and sand content was developed. The model was then implemented in ABAQUS, and its reliability was checked by comparing the numerical data with the triaxial test under the confining pressure of 500 kPa.

## Research significance

Previous experimental studies on sand-fine mixtures, which have been reported in the literature review, indicated sand content has significant effects on the liquefaction resistance and shear strength of sand-fine mixtures. However, no effective model has been developed for quantitatively assessing the import of sand content on the mechanical behavior of sand-fine mixtures. Therefore, the work in this paper can provide a reference for performing theoretical research about the soil.

## Laboratory testing

### Experimental material

The sampling site was approved by Tongxin County Water Affairs Bureau. The soil specimens were collected from the Suocaowozi Reservoir, Tongxin County, Ningxia, China. In order to understand the profile distribution of the particle size for the sediment in Suocaowozi Reservoir, the author counted 36 soil samples (12 samples for each test pit), 30 samples of which were taken every 0.5m within the range of 0-5m, and the rest were taken every 1m within the range of 5-7m. The natural moisture content, natural dry density, and particle size composition of the silted soil are shown in Fig 1 (S1 Table in S1 File) and Table 1.

It can be seen from Fig 1 that the sand distribution of silted soil at different depths mainly ranges from 0% to 40%, but the sand content at 6m is 55.42%. Therefore, the maximum proportion of sand was set as 60%, and the others from high to low in turn were 50%, 28.57%, 16.67%, and 0%. Before preparing the sample, the soil sample shall be air-dried and then sieved into two different ranges of particle sizes: below 0.075 mm and 0.075–2 mm. Soil particles whose particle size ranges from 0.075 to 2 mm belong to sand, and the particle size of fine particles is less than 0.075 mm. All sand and fine particles were mixed uniformly with sand in different proportions to get reconstructed soil specimens with a wide range of sand content ($F_s$ = 0%, 16.67%, 28.57%, 50%, and 60%). The fundamental physical properties of sand-silt mixtures with different sand contents were obtained from particle analysis, specific gravity, liquid limit, and plastic limit tests, as presented in Fig 2 (S2 Table in S1 File) and Table 2.

With a dry density of 1.66 g/cm$^3$ and a moisture content of 22%, cylindrical specimens of 50 mm in diameter and 100 mm in height were created. In order to guarantee uniformity in the remolded samples, each sample was prepared in five layers. For each sand content, four tests were conducted at different confining pressure, and each test was run three times to

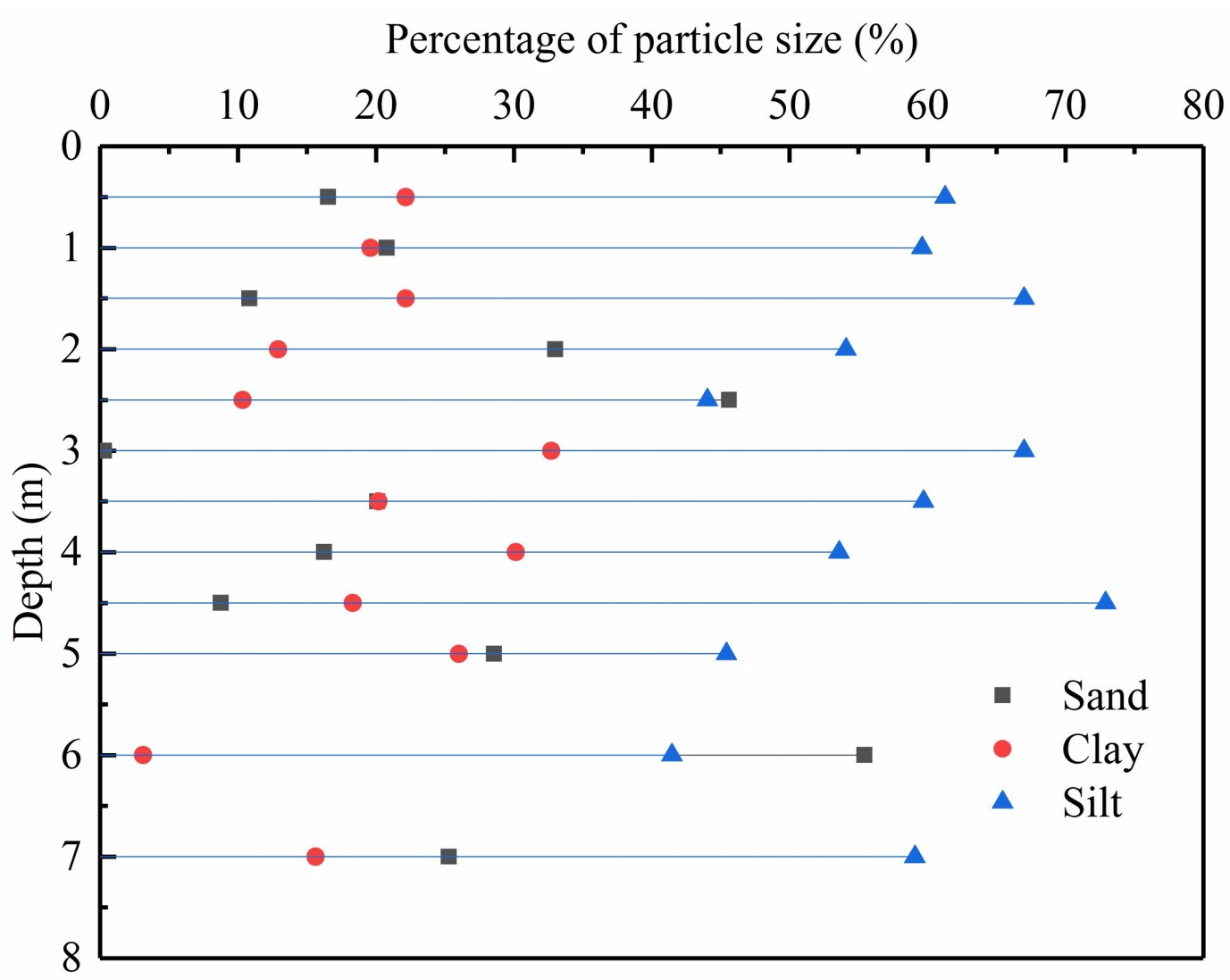

**Fig 1. The particle size profile of silt soil in Suocaowozi Reservoir.**

guarantee the consistency and repeatability of the results. In total, 60 tests were performed (i.e., 5×4×3 = 60).

### Experimental equipment

As indicated in Fig 3, the test apparatus used in this study is an automatic stress-strain path tri-axial apparatus (GSY-SYL-100) that can measure the volumetric strain of the specimen. The displacement of the axial loading piston ('4' in Fig 3), which was measured by the displacement

**Table 1. The natural moisture content and natural dry density of silted soil in Suocaowozi Reservoir.**

| Property | Maximum | Minimum | Average | Sample variance |
|---|---|---|---|---|
| Natural moisture content | 25.32 | 10.45 | 22.05 | 12.30 |
| Natural dry density | 1.67 | 1.62 | 1.66 | 1.63 |

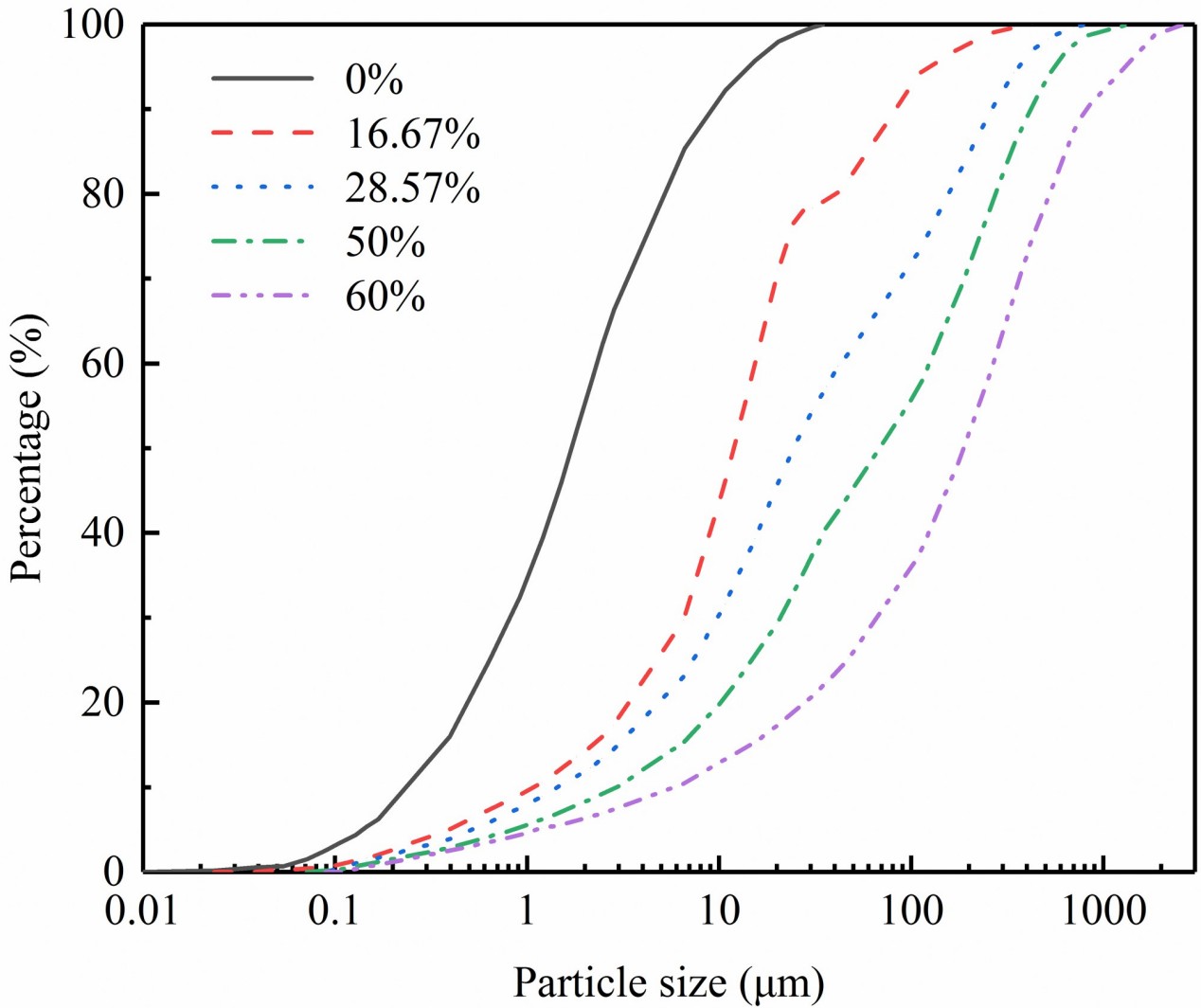

**Fig 2. The gradation curve of sand-silt mixtures with different sand contents.**

transducer ('6' in Fig 3), can be utilized for determining the specimen's axial strain. The volumetric strain of the specimen can be measured by the osmotic actuator('8' in Fig 3).

The apparatus has two controlling options: load control and displacement control. The computer program automatically controls the testing process, and the experimental data is

**Table 2. The basic physical indexes of sand-silt mixtures with different sand contents.**

| Sand content | Specific gravity | Liquid limit | Plastic limit | Plasticity index |
| --- | --- | --- | --- | --- |
| /% | | /% | /% | |
| 0 | 2.72 | 30.1 | 16.1 | 14 |
| 16.67 | 2.67 | 25.7 | 15.2 | 10.5 |
| 28.57 | 2.65 | 23.3 | 13.2 | 10.1 |
| 50 | 2.62 | 19.5 | 11.9 | 7.6 |
| 60 | 2.61 | 18.2 | 11.2 | 7.0 |

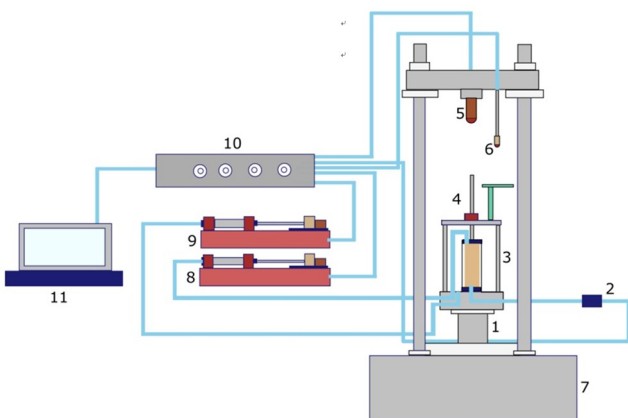

**Fig 3. The schematic diagram of stress-strain triaxial apparatus (GSY-SYL-100).** 1—Lifting table; 2—Pore pressure sensor; 3—Pressure gauge; 4—Piston ejector rod; 5—Pressure sensor; 6—Displacement meter; 7—Testing machine; 8—Osmotic actuator; 9—Confining pressure actuator; 10—Data acquisition box; 11- Computer.

automatically collected. The maximum axial load is 20 kN. The axial displacement and confining pressure of the triaxial apparatus have ranges of –30 cm to 30 cm and 0 MPa to 2 MPa, respectively. Deionized water is utilized as the loading liquid in confining pressure loading system.

## Experimental procedures

A series of CD tests were carried out in three steps: (1) The degree of saturation of the soil samples was raised to >95% using the vacuum method. (2) The soil samples were saturated by applying cycles of cell pressure and backpressure to get a B-value of $\geq$ 0.98 before triaxial tests were performed. (3) Drained shearing was conducted on the soil specimens at a strain rate of 0.005 mm/s. During the shearing process, four separate tests were carried out on the soil specimens with a constant sand content at different confining pressures (i.e., $s_3$ = 100, 200, 300, and 400 kPa) until the axial strain of 25%.

## Testing results

### Stress-strain relationship

Fig 4 (S4 Table in S1 File) provides the test results for all samples in terms of deviatoric stress ($\sigma_1 - \sigma_3$) vs. axial strain ($\varepsilon_1$). The three samples produced fairly consistent results in all situations for each sand content ($F_s$) and confining pressure; As a result, for the sake of simplicity, just the average of the three tests is displayed for each case in Fig 4. On the basis of the test results, the stress-strain curves of sand-silt mixtures with different sand contents behave softening under each confining pressure. Moreover, as the confining pressure rises, the degree of the strain-softening also does as well.

### Peak deviator stress

The peak deviator stress($q_f$) versus $F_s$ curves obtained from the tests are shown in Fig 5 (S5 Table in S1 File). At a specific confining pressure, The change between the peak deviator stress and sand content shows a peak value at a sand content of 28.57%. For sand-silt mixtures with a sand content less than 28.57%, the peak deviator stress is inversely correlated with the

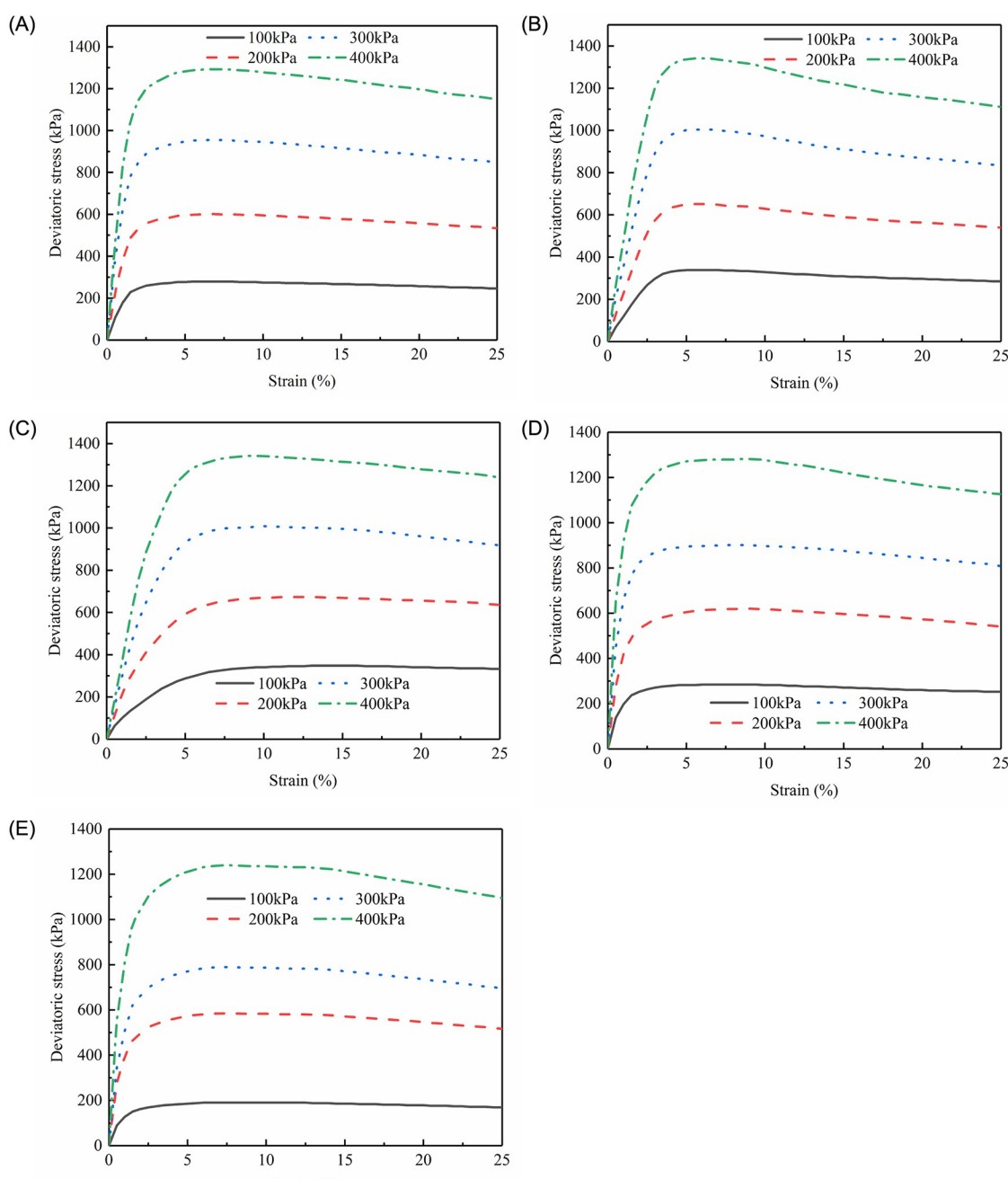

**Fig 4. The stress versus strain curves for samples with different sand contents (A)$F_s$ = 0%; (B) $F_s$ = 16.67%; (C) $F_s$ = 28.57%; (D) $F_s$ = 50%; and (E) $F_s$ = 60%.**

magnitude of sand content, but the tendency is the opposite for higher values of sand content over 28.57%. A quadratic function can be used to fit the relationship between the peak deviator stress ($q_f$) and sand content ($F_s$), as given in Eq 1. The fitting effect and fitting parameters are presented in Fig 6 and Table 3, respectively.

$$q_f = a + bF_s + dF_s{}^2 \tag{1}$$

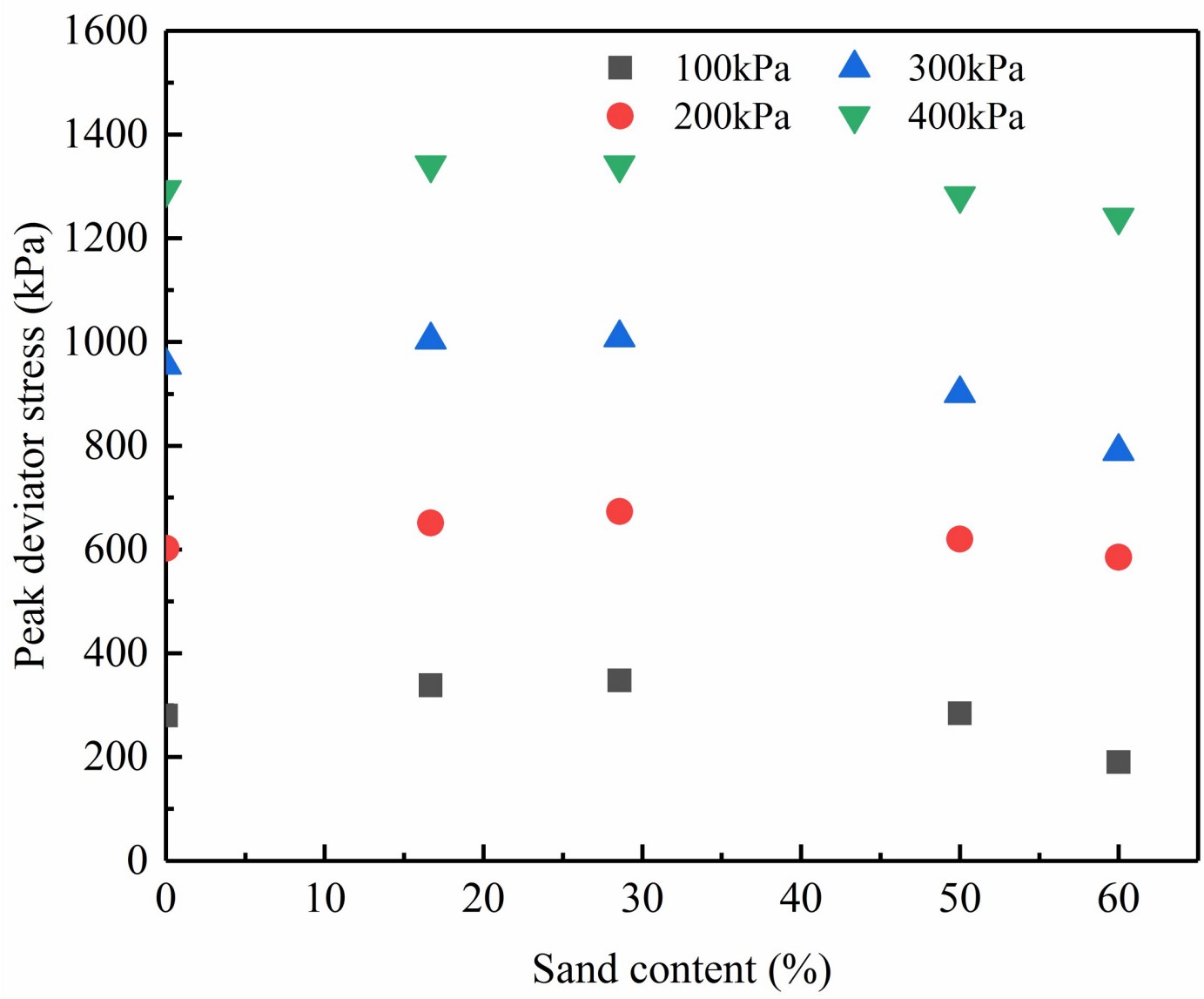

**Fig 5. The relationship between peak deviator stress and sand content.**

where $a$, $b$, and $d$ are the dimensionless parameters. The dimensionless parameters $b$ and $d$ are independent of the confining pressures, as shown in Table 3. By contrast, the dimensionless parameter $a$ varies significantly when the confining pressure changes.

For the test results of this research, the change of parameter $a$ with the confining pressure is illustrated in Fig 7. This figure illustrates that the dependence of parameter $a$ on the confining pressure can be properly represented by a linear relationship (correlation coefficient of $R^2 = 0.99$), which can be written as follows:

$$a = A_1 + A_2\sigma_3 \tag{2}$$

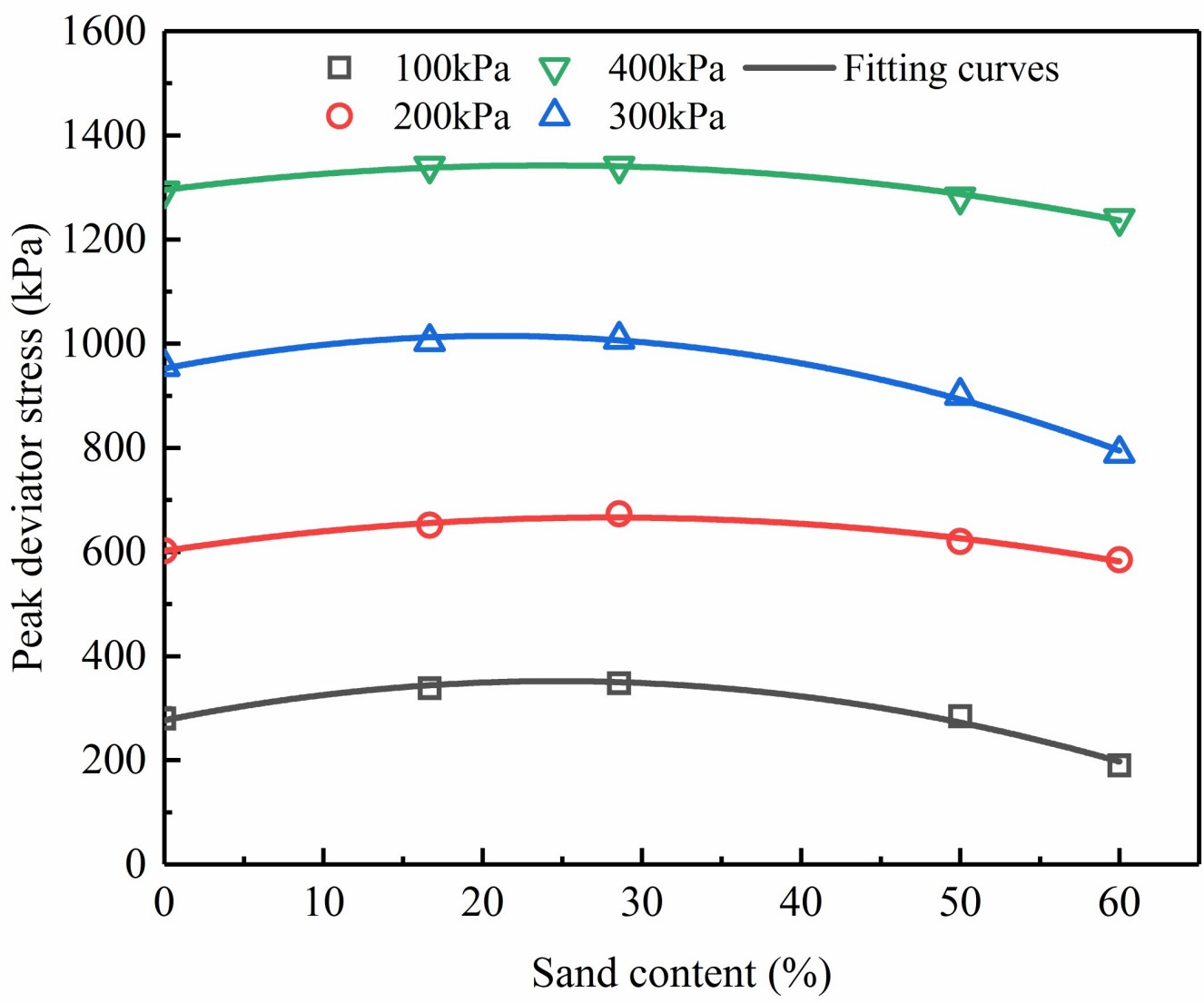

**Fig 6. The relationship between peak deviator stress and sand content.**

The values of the dimensionless parameters $A_1$ and $A_2$ obtained are -69.782 and 3.405, respectively, for the soil of interest in this study.

Combined with Eqs 1 and 2, the relationship between peak deviatoric stress and sand content and confining pressure can be obtained.

$$q_f = 3.405\sigma_3 - 69.782 + bF_s - dF_s^2 \tag{3}$$

## Volumetric strain–axial strain relationship

The relationships between the volumetric strain ($\varepsilon_v$) and the axial strain ($\varepsilon_a$) under various confining pressures are shown in Fig 8. Because of the observed repeatability of the test data, the average results of the three tests are offered. At the same fine content, the slope of the curves gradually rises in absolute value (the slopes are negative) as the confining pressure

**Table 3. The fitting results of peak deviatoric stress and sand content.**

| Confining pressure /kPa | *a* | *b* | *d* | $R^2$ |
|---|---|---|---|---|
| 100 | 276.72 | 6.097 | -0.124 | 0.97 |
| 200 | 601.931 | 4.587 | -0.082 | 0.95 |
| 300 | 951.988 | 6.011 | -0.144 | 0.98 |
| 400 | 1294.935 | 3.930 | -0.084 | 0.98 |

increases during the initial shear contraction. Following the cessation of contractions, all samples display dilative behavior since they shear toward failure. At an axial strain of about 5% in each test, the volume strain vs. axial strain curves reaches their extreme value, referred to as the maximum shrinkage.

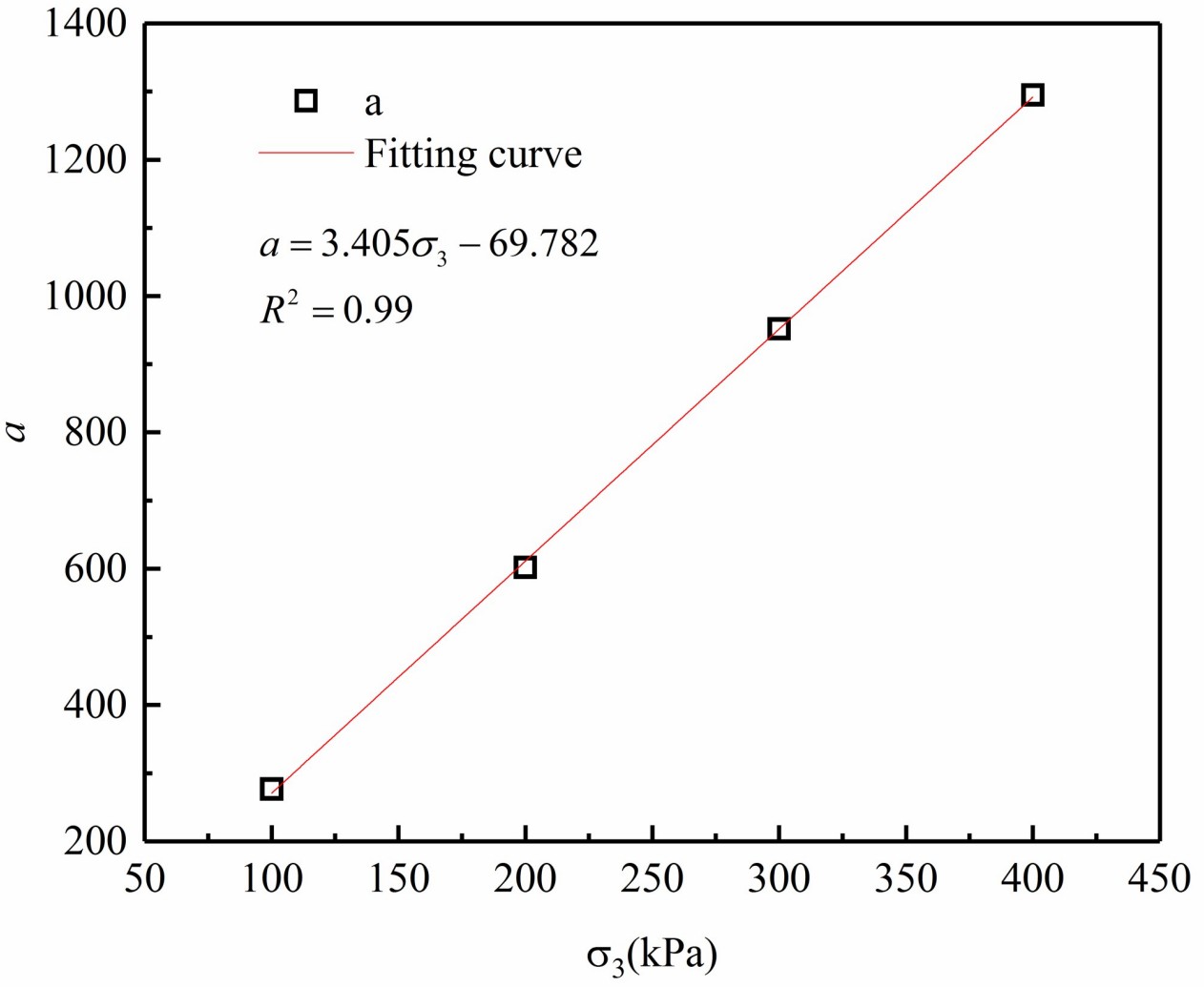

**Fig 7. The fitting relationship between *a* and confining pressure $\sigma_3$.**

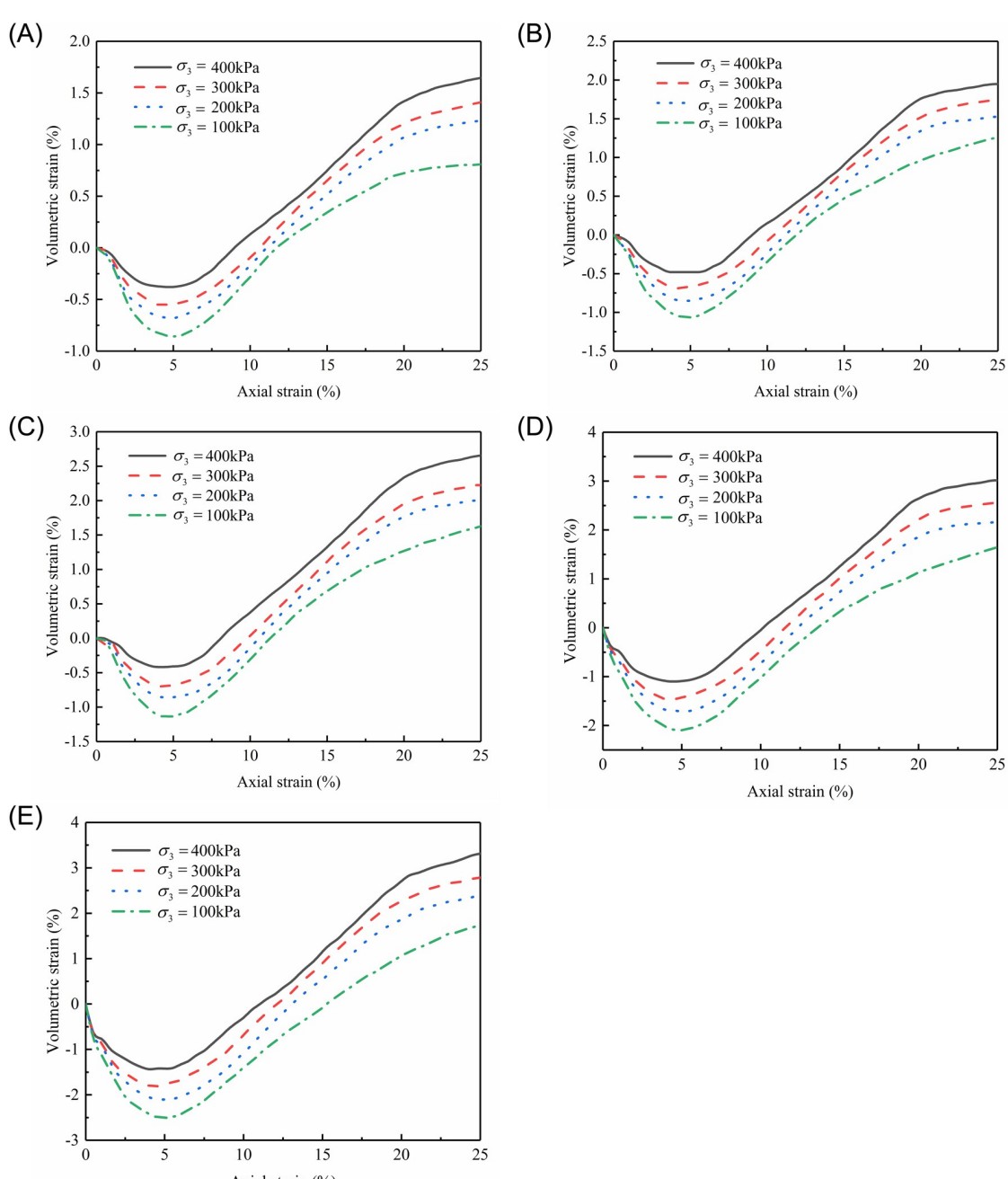

**Fig 8. Relationships between volumetric strain and axial strain:** (A) $F_s$ = 60%; (B) $F_s$ = 50%; (C) $F_s$ = 28.57%; (D) $F_s$ = 16.67%; (E) $F_s$ = 0%.

## Modified Duncan-Chang model

### Modified Duncan-Chang model

The test results show that the $\sigma_1 - \sigma_3$ verses $\varepsilon_1$ curves of sand-silt mixtures with different sand contents behave with strain softening. However, The Duncan-Chang model can only capture the strain-hardening characteristics of the tested soil [21]. To better represent the stress-strain

response of sand-silt mixtures, the Duncan-Chang model needs to be improved as follows:

$$q = \sigma_1 - \sigma_3 = \frac{\varepsilon_1}{A + B\varepsilon_1 + C\sqrt{\varepsilon_1}} \tag{4}$$

where $A$, $B$, and $C$ are the model parameters. The test shows that the parameter $C$ is usually negative.

By differentiating Eq 4, the following equation can be obtained:

$$\frac{d(\sigma_1 - \sigma_3)}{d\varepsilon_1} = \frac{C\varepsilon_1 + 2A\sqrt{\varepsilon_1}}{2\sqrt{\varepsilon_1}\left(A + B\varepsilon_1 + C\sqrt{\varepsilon_1}\right)^2} \tag{5}$$

$\frac{d(\sigma_1-\sigma_3)}{d\varepsilon_1}$ is an increasing function when $\varepsilon_1 < \frac{4A^2}{C^2}$; while $\frac{d(\sigma_1-\sigma_3)}{d\varepsilon_1}$ is a decreasing function when $\varepsilon_1 > \frac{4A^2}{C^2}$. Hence, strain-softening behavior can be characterized using Eq 4. When $C = 0$, Eq 4 is the mathematical expression of the Duncan-Chang model used to approximate the strain-hardening stress-strain curves.

For convenience, the expressions of the model parameters $A$, $B$, and $C$ are listed in the paper.

$$\left.\begin{array}{l} A = \dfrac{1}{E_0} \\[2mm] B = \dfrac{1}{q_r} \\[2mm] C = \pm\dfrac{2}{E_0\sqrt{\varepsilon_{af}}} \end{array}\right\} \tag{6}$$

where $E_0$ is the initial tangent modulus, $q_r$ is the residual deviatoric stress, $\varepsilon_{af}$ is the axial strain that corresponds to the peak deviator stress $q_f$, as shown in Fig 9.

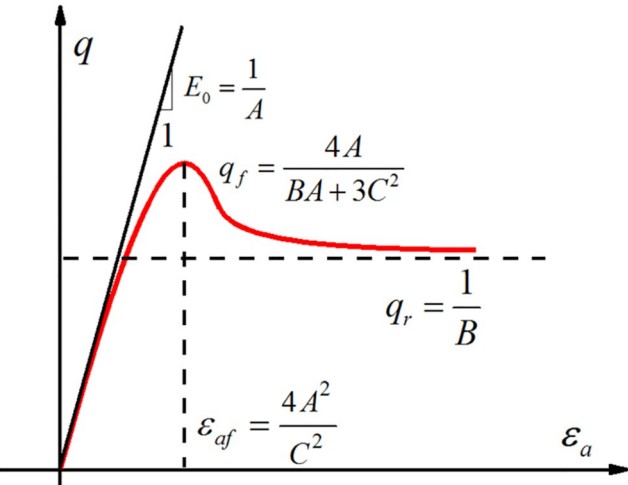

**Fig 9. The physical meaning of the model parameters.**

## Verification on coral clay

On marine coral clay samples with an initial dry density of 1.3 g/cm³ and water content of 10%, Jiang et al. [22] performed consolidated undrained triaxial tests. Experimental results demonstrate a softening of the sample's stress-strain relationship. In this paper, under confining pressures of 100, 200, 300, and 400 kPa, the stress-strain relationship is described by the modified Duncan-Chang model and the initial Duncan-Chang model, respectively. Comparison results between model calculation values and experimental values are shown in Fig 10 (S10 Table in S1 File), and relevant parameters are summarized in Table 4. It can be seen that the calculated values of the modified Duncan-Chang model have good consistency with the testing values, indicating that the modified Duncan-Chang model can accurately reflect the softening stress-strain curves of coral clay. However, there exists a big difference between the predicted stress-strain curves from the initial Duncan-Chang model and the measured stress-strain curves.

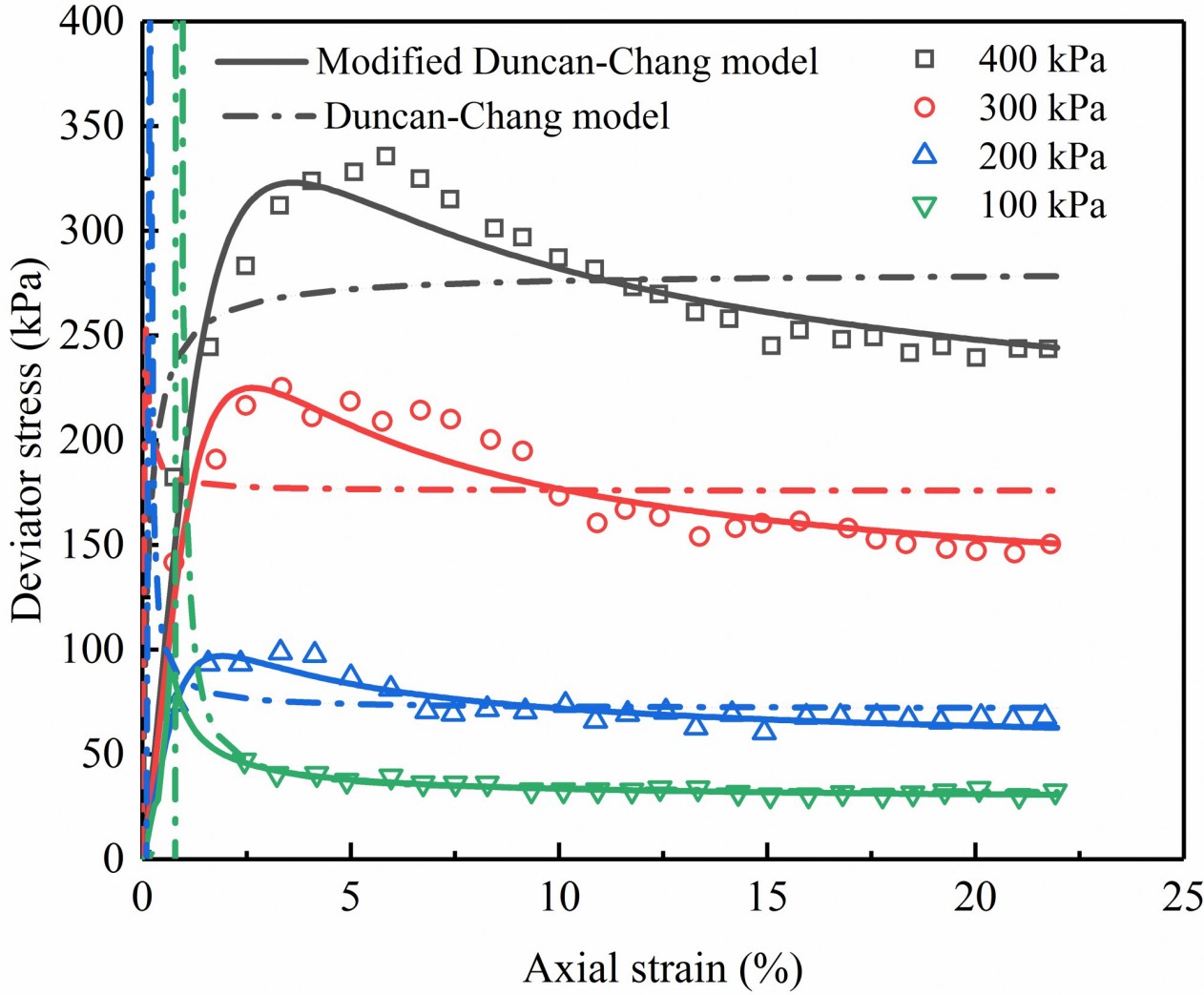

**Fig 10. Comparison of experimental data and model stress-strain relationships of coral clay.**

**Table 4. Model parameter values by the modified Duncan-Chang model for the coral clay.**

| Confining pressure / kPa | *A* | *B* | *C* |
|---|---|---|---|
| 100 | 0.01027 | 0.005923 | -0.01078 |
| 200 | 0.01367 | 0.009577 | -0.01676 |
| 300 | 0.022 | 0.02161 | -0.03157 |
| 400 | 0.00517 | 0.03839 | -0.02969 |

## Verification on loess

Wang et al. [14] carried out a number of drained triaxial compression tests on undisturbed loess under the confining pressures of 50, 200, 300, and 500 kPa. The stress-strain curve of undisturbed loess is softening under the confining pressures of 50 kPa and 200 kPa, and hardening under the confining pressures of 300 kPa and 500 kPa. In order to verify the modified Duncan-Chang in the study, the triaxial tests performed on the specimens under the confining pressure of 50 kPa and 200 kPa were simulated, as shown in Fig 11 (S11 Table in S1 File), and the parameters are listed in Table 5. To compare with the initial Duncan Chang model, the curves described by the initial Duncan Chang model are also presented in Fig 11.

Under the confining pressure of 50 kPa, the described result calculated from the modified Duncan-Chang model is highly consistent with the measured stress-strain relationship compared to the initial Duncan-Chang model, demonstrating the modified Duncan-Chang model can correctly describe the stress-strain relationship of undisturbed loess. From the development trend of the test curves under the confining pressure of 200 kPa, the stress-strain curves predicted by the modified Duncan-Chang model and initial Duncan-Chang model coincide with the experimental data, but the results of the modified Duncan-Chang model are more accurate than that of the initial Duncan-Chang model.

## Influence of sand content on initial tangent modulus

According to the definition of initial tangent modulus, the following equation can be obtained:

$$E_0 = \frac{d(\sigma_1 - \sigma_3)}{d\varepsilon_1}\bigg|_{\varepsilon_1=0} = \frac{1}{A} \tag{7}$$

Numerous academic studies propose that the confining pressure and the soil's initial tangent modulus are related in an exponential fashion as follows:

$$E_0 = K p_a \left(\frac{\sigma_3}{p_a}\right)^n \tag{8}$$

where $p_a$ is the reference atmospheric pressure (approximately 100 kPa), $K$ is a dimensionless modulus number, and $n$ is a dimensionless modulus exponent.

Considering this format, if $\lg(E_0/p_a)$ versus $\lg(\sigma_3/p_a)$ is plotted a linear relationship with the slope of $n$ and $y$-intersection of $\lg K$ is expected

$$\lg\left(\frac{E_0}{p_a}\right) = \lg K + n\lg\left(\frac{\sigma_3}{p_a}\right) \tag{9}$$

These plot types are created for the test results of this investigation, as demonstrated in Fig 12. This chart illustrates the linearity of the relationships, which have a correlation coefficient

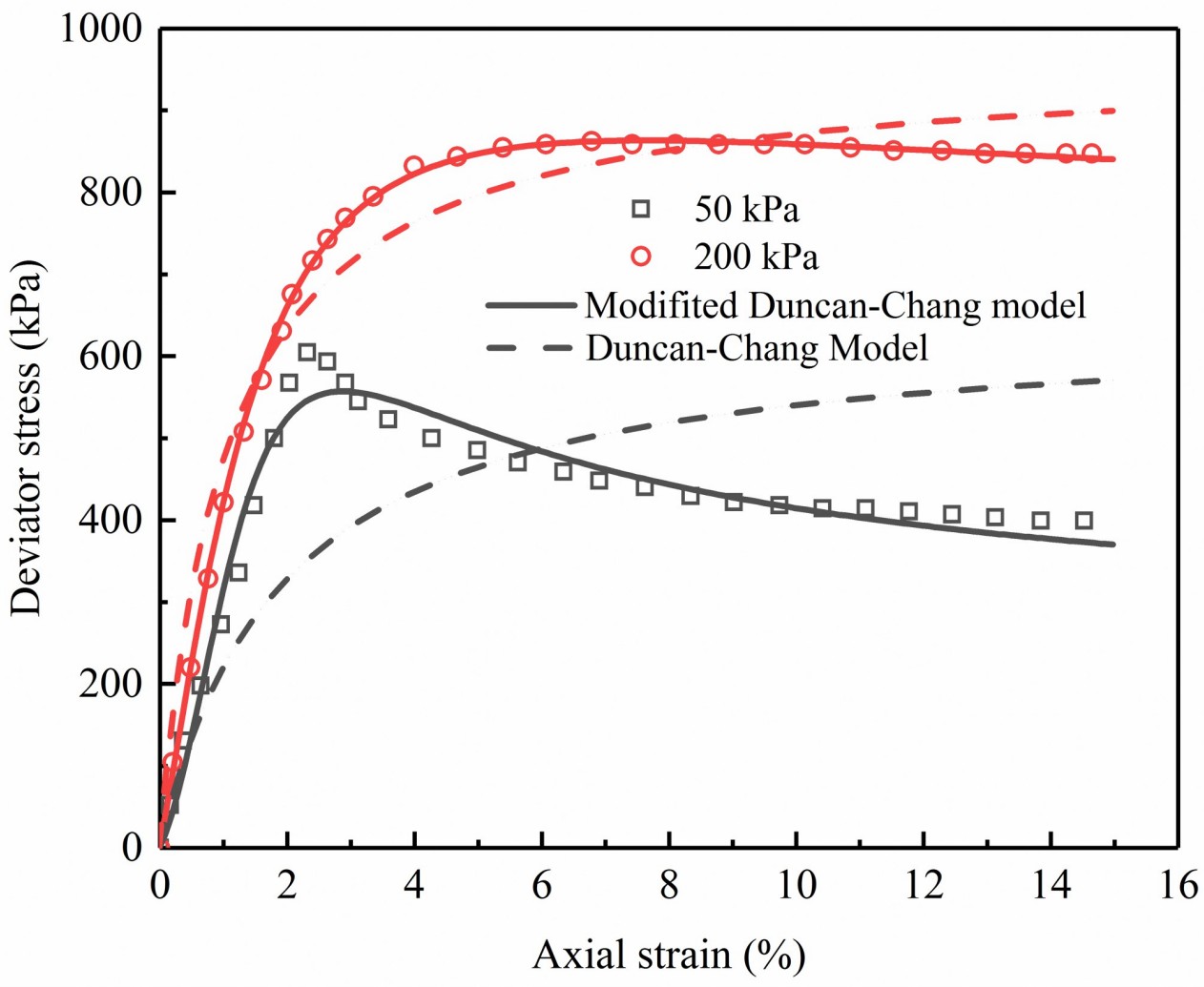

**Fig 11. Comparison of experimental data and model stress-strain relationships of undisturbed loess.**

($R^2$) of greater than 94%. As shown in Table 6, the parameters $K$ and $n$ are determined appropriately for various sand contents.

According to Table 6, the parameters $K$ and $n$ are dependent on the sand content. The relation of fitting between $K$ and $n$ and sand content are respectively shown in Figs 13 and 14 and are presented as follows:

$$K = \begin{cases} k_1 + k_2 F_s + k_3 F_s^2 & \text{for } F_s < 28.57\% \\ k_4 + k_5 F_s + k_6 F_s^2 & \text{for } F_s \geqslant 28.57\% \end{cases} \tag{10}$$

**Table 5. Described parameter values for undisturbed loess.**

| Confining pressure / kPa | A /$10^{-3}$ | B /$10^{-3}$ | C /$10^{-3}$ |
|---|---|---|---|
| 50 | 8.277 | 4.66 | -9.743 |
| 200 | 3.007 | 1.549 | -2.17 |

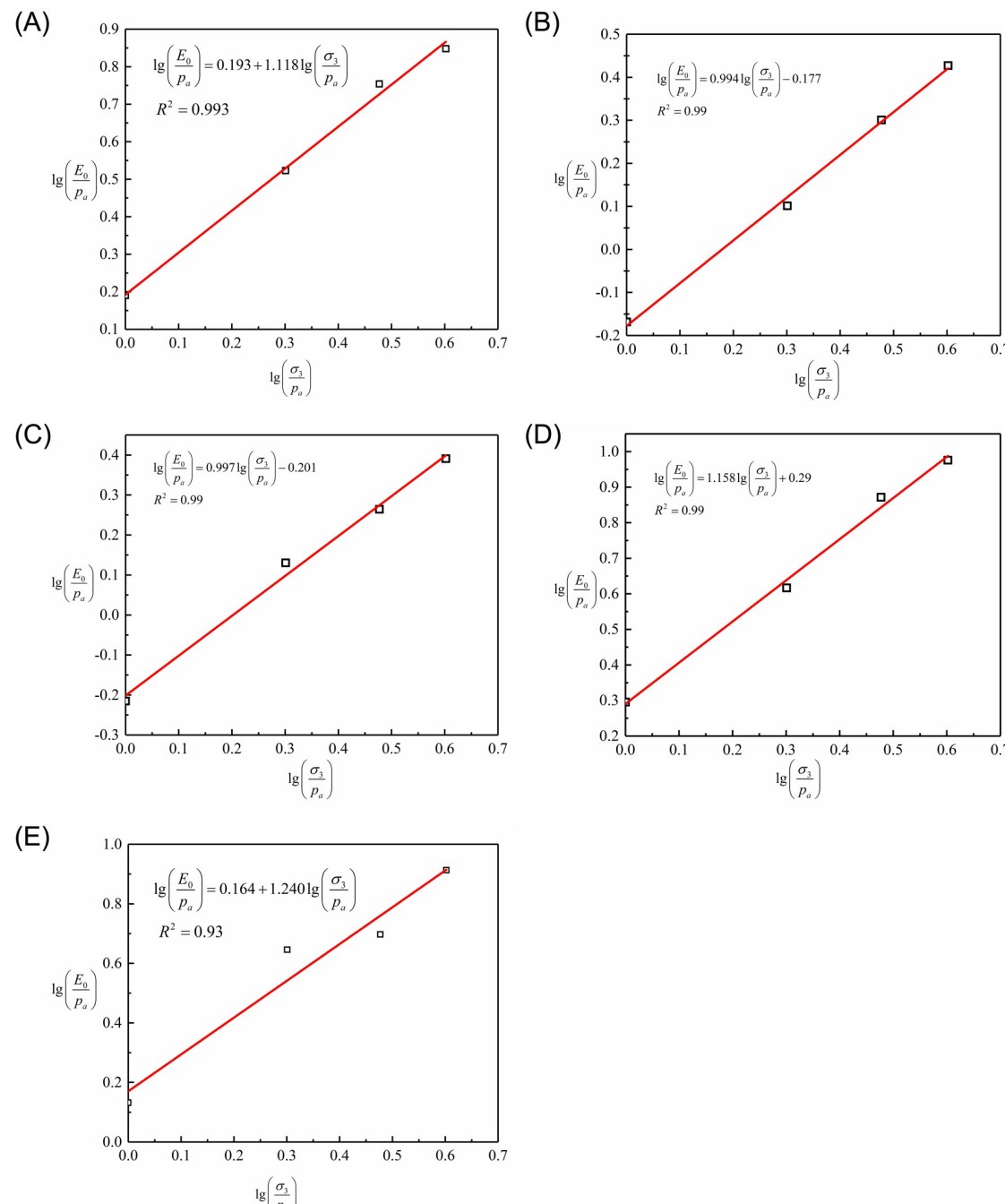

**Fig 12. Lg($E_0/p_a$) versus lg($\sigma_3/p_a$) obtained from the CD tests with different sand contents: (A)$F_s$ = 0%; (B) $F_s$ = 16.67%; (C) $F_s$ = 28.57%;(D) $F_s$ = 50%; (E)$F_s$ = 6 0%.**

$$n = n_1 + n_2 F_s + n_3 F_s^2 \qquad (11)$$

Fig 13 illustrates how Eq 10 represents the dependence of $K$ to $F_s$, with the correlation coefficient>99.6%. Likewise, it can be seen from Fig 14 that Eq 11 can accurately depict the dependency of $n$ on the sand content, and the corresponding correlation

**Table 6. Values of $K$ and $n$ for different sand contents.**

| $F_s$ /% | $\sigma_3$ /kPa | lg $(\sigma_3 / p_a)$ | lg $(E_0 / p_a)$ | $K$ | $n$ |
|---|---|---|---|---|---|
| 0 | 100 | 0.000 | 0.190 | 1.557 | 1.119 |
| | 200 | 0.301 | 0.523 | | |
| | 300 | 0.477 | 0.754 | | |
| | 400 | 0.602 | 0.848 | | |
| 16.67 | 100 | 0.000 | 0.262 | 0.664 | 0.994 |
| | 200 | 0.301 | 0.589 | | |
| | 300 | 0.477 | 0.795 | | |
| | 400 | 0.602 | 0.880 | | |
| 28.57 | 100 | 0.000 | 0.456 | 1.589 | 0.997 |
| | 200 | 0.301 | 0.683 | | |
| | 300 | 0.477 | 0.840 | | |
| | 400 | 0.602 | 0.985 | | |
| 50 | 100 | 0.000 | 0.295 | 1.951 | 1.158 |
| | 200 | 0.301 | 0.634 | | |
| | 300 | 0.477 | 0.815 | | |
| | 400 | 0.602 | 0.976 | | |
| 60 | 100 | 0.000 | 0.132 | 1.459 | 1.240 |
| | 200 | 0.301 | 0.646 | | |
| | 300 | 0.477 | 0.697 | | |
| | 400 | 0.602 | 0.913 | | |

coefficient = 99.9%. For the sand-silt mixtures of this study, the values of dimensionless parameters $k_1$, $k_2$, $k_3$, $k_4$, $k_5$, $k_6$, $n_1$, $n_2$, and $n_3$ obtained are 1.543, -0.129, 0.004, -1.878, 0.180, 0.002, 1.111, -0.009, and 1.995, respectively, as presented in Figs 13 and 14.

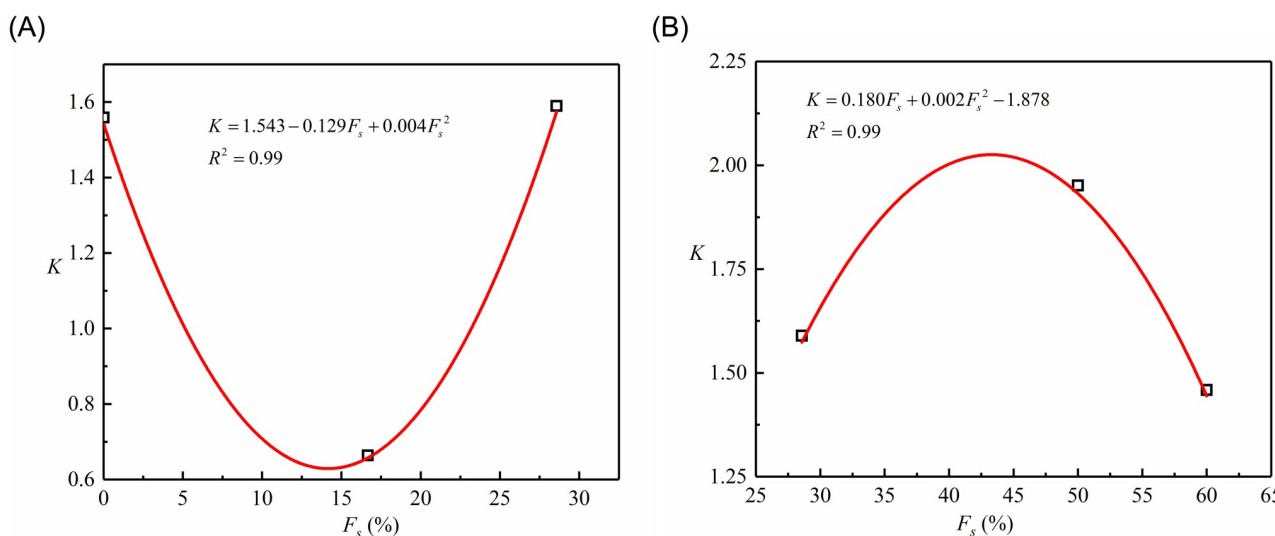

**Fig 13. Fitted correlation between $K$ and sand content: (A) $F_s<$28.57%; (B) $F_s\geqslant$28.57%.**

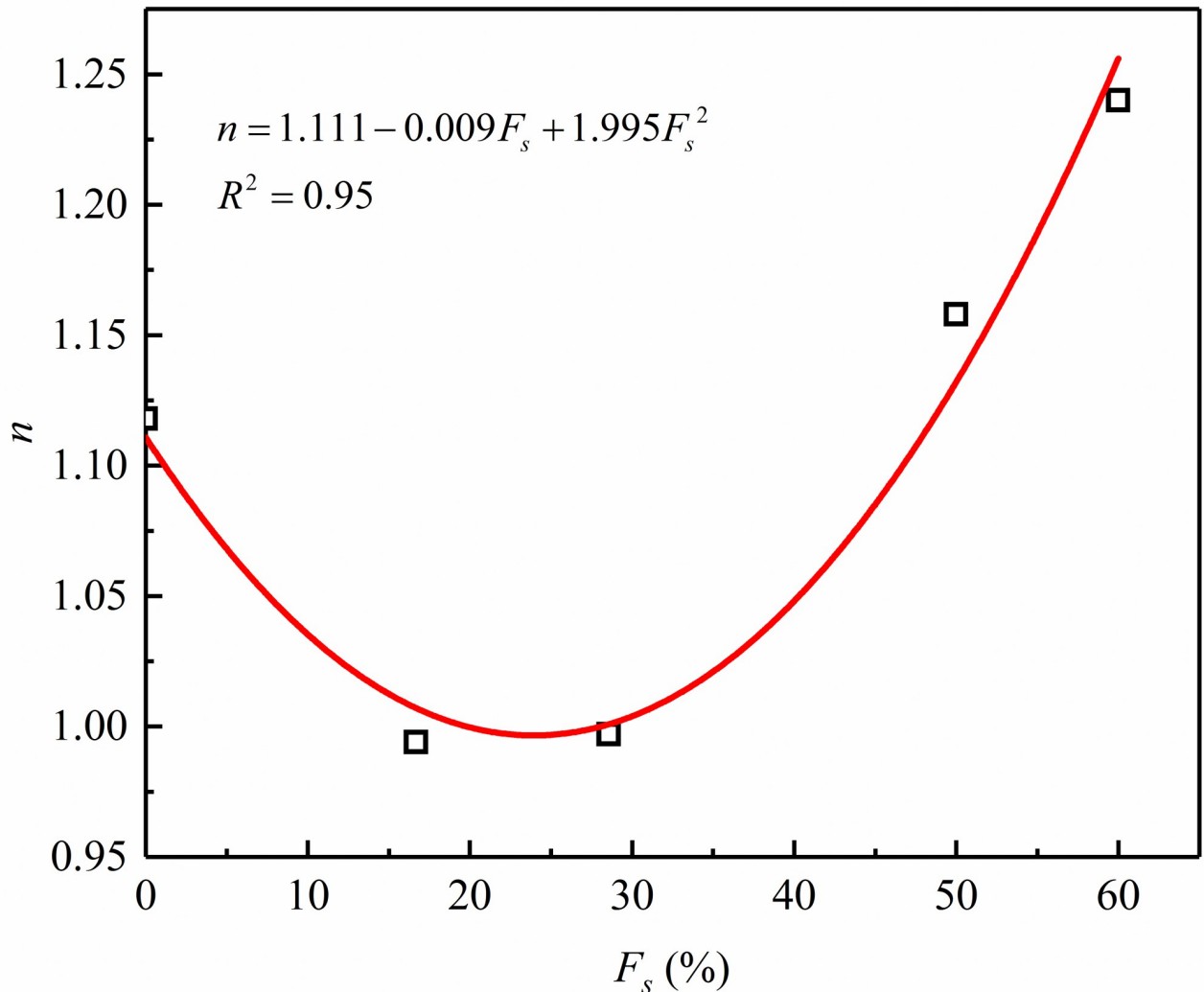

**Fig 14. Fitted correlation between _n_ and sand content.**

According to Fig 8, the relationship between axial strain and volumetric strain can be represented by a cubic function:

$$\varepsilon_1 = M_1 \varepsilon_v^{\ 3} + M_2 \varepsilon_v^{\ 2} + M_3 \varepsilon_v \tag{12}$$

According to the definition of tangent Poisson's ratio, the following equation can be obtained:

$$\upsilon_t = \frac{\partial \varepsilon_3}{\partial \varepsilon_1} = \frac{-\varepsilon_v (M_1 \varepsilon_v^{\ 2} + M_2 \varepsilon_v + M_3 - 1)}{6 M_1 \varepsilon_v^{\ 2} + 4 M_2 \varepsilon_v + 2 M_3} \tag{13}$$

where $M_1$, $M_2$, and $M_3$ are the material constants related to confining pressure ($\sigma_3$), and can be determined by fitting the experimental data, as shown in Fig 15 and Table 7.

As shown in Table 7, the parameters $M_1$, $M_2$, and $M_3$ are dependent on confining pressure ($\sigma_3$). The relation of fitting between $M_1$, $M_2$, and $M_3$ and confining pressure are presented in Fig 16. The relations of $M_1$, $M_2$, $M_3$, and $\sigma_3$ are approximately linearity and can be expressed

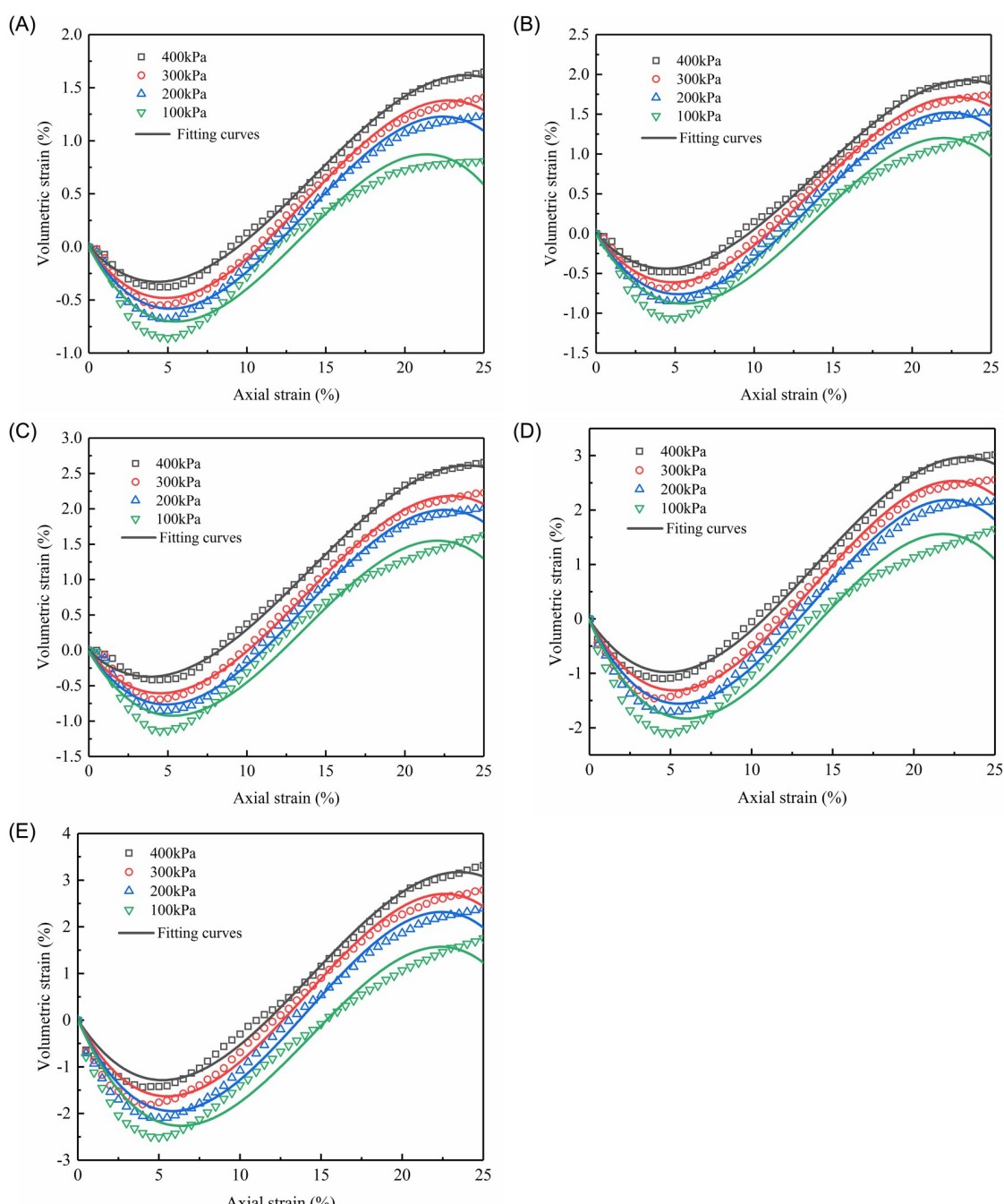

**Fig 15. The axial strain versus volumetric strain for sand-silt mixtures with different sand contents: (A)** $F_s$ = 60%; **(B)** $F_s$ = 50%; **(C)** $F_s$ = 28.57%; **(D)** $F_s$ = 16.67%;**(E)** $F_s$ = 0%.

by Eqs 14–16.

$$M_1 = M_{1a} + M_{1b}\sigma_3 \tag{14}$$

$$M_2 = M_{2a} + M_{2b}\sigma_3 \tag{15}$$

**Table 7. Values of $M_1$, $M_2$, and $M_3$.**

| $F_s$ | $\sigma_3$ | $M_1$ | $M_2$ | $M_3$ |
|---|---|---|---|---|
| /% | /kPa | | | |
| 0 | 100 | -5.209E-4 | 0.02208 | -0.1627 |
| | 200 | -6.438E-4 | 0.02674 | -0.2147 |
| | 300 | -7.037E-4 | 0.02915 | -0.2453 |
| | 400 | -7.887E-4 | 0.03185 | -0.2799 |
| 16.67 | 100 | -6.805E-4 | 0.02843 | -0.2100 |
| | 200 | -8.062E-4 | 0.03342 | -0.2674 |
| | 300 | -9.032E-4 | 0.03731 | -0.3143 |
| | 400 | -9.362E-4 | 0.03868 | -0.3429 |
| 28.57 | 100 | -7.247E-4 | 0.0303 | -0.2006 |
| | 200 | -8.955E-4 | 0.03694 | -0.2807 |
| | 300 | -9.981E-4 | 0.04115 | -0.332 |
| | 400 | -0.00105 | 0.04333 | -0.3728 |
| 50 | 100 | -0.0013 | 0.05374 | -0.4319 |
| | 200 | -0.0015 | 0.06261 | -0.5371 |
| | 300 | -0.0016 | 0.06802 | -0.6077 |
| | 400 | -0.0017 | 0.07114 | -0.6704 |
| 60 | 100 | -0.0014 | 0.0625 | -0.5323 |
| | 200 | -0.0017 | 0.0728 | -0.6444 |
| | 300 | -0.0018 | 0.0789 | -0.7308 |
| | 400 | -0.0020 | 0.0910 | -0.7887 |

$$M_3 = M_{3a} + M_{3b}\sigma_3 \tag{16}$$

where $M_{1a}$, $M_{1b}$, $M_{2a}$, $M_{2b}$, $M_{3a}$, and $M_{3b}$ are the material constants related to sand content ($F_s$), as shown in Fig 17 (S17 Table in S1 File).

As shown in the above figure, $M_{1a}$, $M_{1b}$, $M_{2a}$, $M_{2b}$, $M_{3a}$, and $M_{3b}$ are the exponential function of sand content and can be expressed by Eqs 17–22.

$$M_{1a} = M_{1a1} \exp\left(\frac{-F_s}{M_{1a2}}\right) + M_{1a3} \tag{17}$$

$$M_{1b} = M_{1b1} \exp\left(\frac{-F_s}{M_{1b2}}\right) + M_{1b3} \tag{18}$$

$$M_{2a} = M_{2a1} \exp\left(\frac{-F_s}{M_{2a2}}\right) + M_{2a3} \tag{19}$$

$$M_{2b} = M_{2b1} \exp\left(\frac{-F_s}{M_{2b2}}\right) + M_{2b3} \tag{20}$$

$$M_{3a} = M_{3a1} \exp\left(\frac{-F_s}{M_{3a2}}\right) + M_{3a3} \tag{21}$$

$$M_{3b} = M_{3b1} \exp\left(\frac{-F_s}{M_{3b2}}\right) + M_{3b3} \tag{22}$$

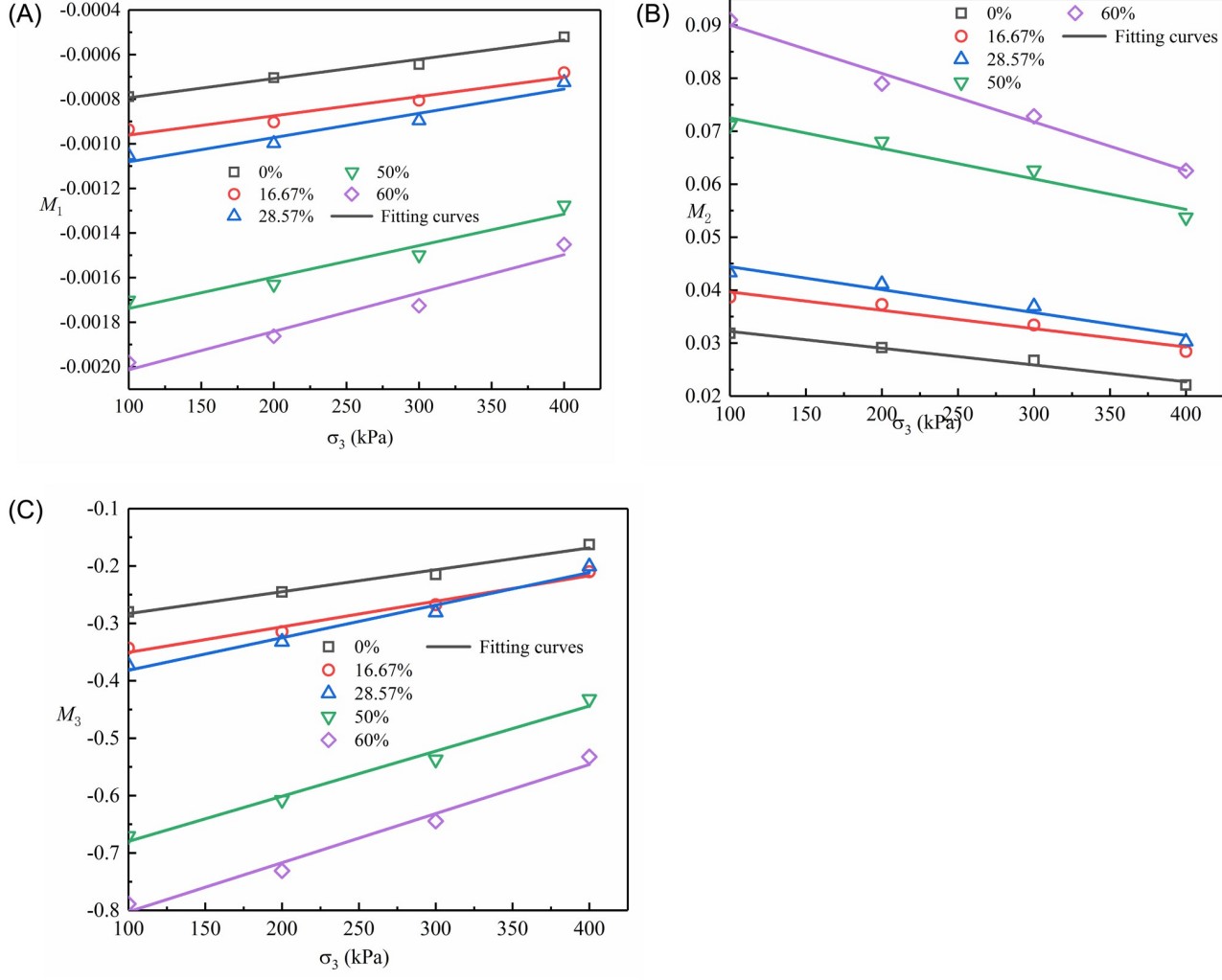

**Fig 16. Fitted correlation between $M_1$, $M_2$, and $M_3$ and confining pressure ($\sigma_3$): (A) $M_1$ vs.$\sigma_3$; (B) $M_2$ vs.$\sigma_3$; (C) $M_3$ vs.$\sigma_3$.**

where $M_{1a1}$, $M_{1a2}$, $M_{1a3}$, $M_{1b1}$, $M_{1b2}$, $M_{1b3}$, $M_{2a1}$, $M_{2a2}$, $M_{2a3}$, $M_{2b1}$, $M_{2b2}$, $M_{2b3}$, $M_{3a1}$, $M_{3a2}$, $M_{3a3}$, $M_{3b1}$ $M_{3b2}$, and $M_{3b3}$ are the material constants and can be determined by fitting experimental data.

Substituting Eqs 17–22 into Eqs 14–16 yields

$$M_{1s} = M_{1a1} \exp(\frac{-F_s}{M_{1a2}}) + M_{1a3} + \left( M_{1b1} \exp(\frac{-F_s}{M_{1b2}}) + M_{1b3} \right)\sigma_3 \tag{23}$$

$$M_{2s} = M_{2a1} \exp(\frac{-F_s}{M_{2a2}}) + M_{2a3} + \left( M_{2b1} \exp(\frac{-F_s}{M_{2b2}}) + M_{2b3} \right)\sigma_3 \tag{24}$$

$$M_{3s} = M_{3a1} \exp(\frac{-F_s}{M_{2a2}}) + M_{3a3} + \left( M_{3b1} \exp(\frac{-F_s}{M_{3b2}}) + M_{3b3} \right)\sigma_3 \tag{25}$$

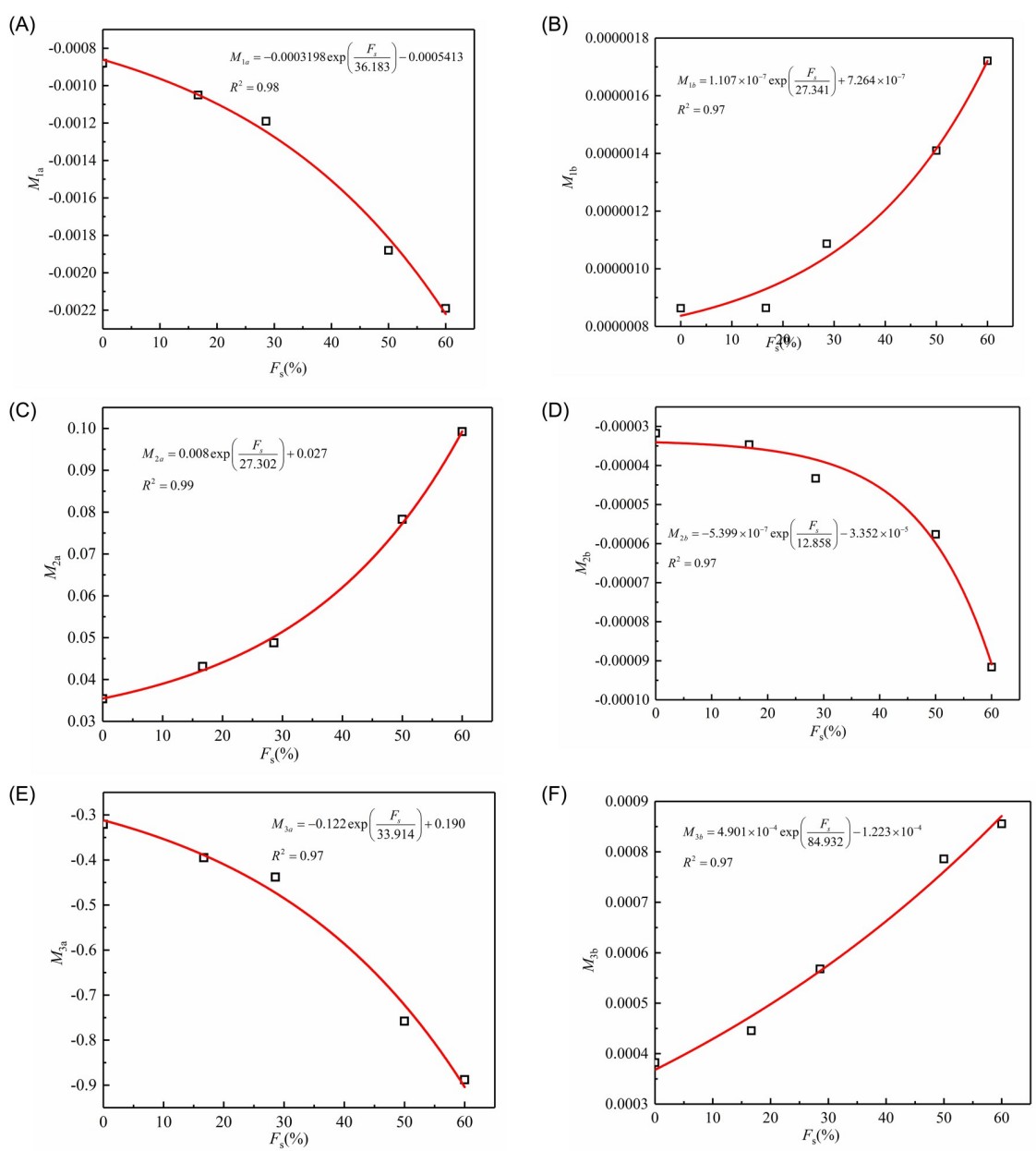

**Fig 17. Fitted correlation between $M_{1a}$, $M_{1b}$, $M_{2a}$, $M_{2b}$, $M_{3a}$, and $M_{3b}$ and $F_s$: (A) $M_{1a}$ vs. $F_s$; (B) $M_{1b}$ vs. $F_s$; (C) $M_{2a}$ vs. $F_s$; (D) $M_{2b}$ vs. $F_s$; (E) $M_{3a}$ vs. $F_s$; (F) $M_{3b}$ vs. $F_s$.**

## Sand-content-dependent constitutive model

According to the definition of the failure ratio in the Duncan-Chang model, the new failure ratio ($R_f$) is defined as follows:

$$R_f = \frac{(\sigma_1 - \sigma_3)_f}{(\sigma_1 - \sigma_3)_r} \tag{26}$$

Therefore,

$$B = \frac{R_f}{(\sigma_1 - \sigma_3)_f} \tag{27}$$

Substituting Eqs 10 and 11 into Eq 8 yields

$$E_s = \begin{cases} (k_1 + k_2 F_s + k_3 F^2) p_a \left(\dfrac{\sigma_3}{p_a}\right)^{n_1 + n_2 F_s + n_3 F_s^2} & \text{for } F_s < 28.57\% \\[3mm] (k_4 + k_5 F_s + k_6 F^2) p_a \left(\dfrac{\sigma_3}{p_a}\right)^{n_1 + n_2 F_s + n_3 F_s^2} & \text{for } F_s \geqslant 28.57\% \end{cases} \tag{28}$$

Substituting Eqs 1 and 2 into Eq 27 yields

$$B_s = \frac{R_f}{A_1 + A_2 \sigma_3 + b F_s + d F_s^2} \tag{29}$$

Then, the modified Duncan-Chang model that takes into account the sand content is produced by substituting Eqs 28 and 29 into Eq 4 as follows:

$$\sigma_1 - \sigma_3 = \frac{\varepsilon_1}{\frac{1}{E_s} + B_s \varepsilon_1 - \frac{2\sqrt{\varepsilon_1}}{E_s \sqrt{\varepsilon_{af}}}} \tag{30}$$

By differentiating Eq 30, the tangent modulus ($E_t$) can be determined:

$$E_t = \frac{E_s \sqrt{\varepsilon_{af}} \left(\sqrt{\varepsilon_{af} \varepsilon_1} - \varepsilon_1\right)}{\sqrt{\varepsilon_1} \left(B_s E_s \varepsilon_1 \sqrt{\varepsilon_{af}} + \sqrt{\varepsilon_{af}} - 2\sqrt{\varepsilon_1}\right)^2} \tag{31}$$

When the deviatoric stress is less than the maximum deviatoric stress in history, the resilience modulus ($E_{ur}$) is accepted:

$$E_{ur} = K_{ur} p_a \left(\frac{\sigma_3}{p_a}\right)^{n_{ur}} \tag{32}$$

where $K_{ur}$ is the unloading-reloading elastic coefficient, which is acquired by utilizing

$$K_{ur} = A_{ur} K \tag{33}$$

where $A_{ur}$ is the unloading ratio from 1.1 to 1.3 [21].

Substituting Eqs 23–25 into Eq 13, tangent Poisson's ratio can be rewritten as:

$$v_t = \frac{-\varepsilon_v (M_{1s} \varepsilon_v^2 + M_{2s} \varepsilon_v + M_{3s} - 1)}{6 M_{1s} \varepsilon_v^2 + 4 M_{2s} \varepsilon_v + 2 M_{3s}} \tag{34}$$

Given the specimen's axial symmetry, the stress tensor and strain tensor for CD tests are respectively expressed as:

$$d\boldsymbol{\sigma} = \left(d\sigma_\rho, d\sigma_z, d\sigma_{\rho z}, d\sigma_\theta\right)^T \tag{35}$$

$$d\boldsymbol{\varepsilon} = \left(d\varepsilon_\rho, d\varepsilon_z, d\varepsilon_{\rho z}, d\varepsilon_\theta\right)^T \tag{36}$$

The constitutive matrix ($D^e$) of the modified Duncan-Chang model at each stress level can be obtained as follows (square brackets denote vectors and matrices):

$$D^e = \frac{E_t(1-v_t)}{(1+v_t)(1-2v_t)} \begin{bmatrix} 1 & \dfrac{v_t}{1-v_t} & 0 & \dfrac{v_t}{1-v_t} \\ \dfrac{v_t}{1-v_t} & 1 & 0 & \dfrac{v_t}{1-v_t} \\ 0 & 0 & \dfrac{1-2v_t}{2(1-v_t)} & 0 \\ \dfrac{v_t}{1-v_t} & \dfrac{v_t}{1-v_t} & 0 & 1 \end{bmatrix} \tag{37}$$

Consequently, the incremental form of the modified Duncan-Chang model can be obtained.

$$\mathrm{d}\dot{\boldsymbol{\sigma}} = D^e \mathrm{d}\boldsymbol{\varepsilon} \tag{38}$$

## Model implementation and verification

### Model implementation

The provided modified Duncan-Chang model is transformed into a user-defined material model (UMAT) subroutine and implemented in ABAQUS (version 6.14). The calculation chart is given in Fig 18. In the model implementation process, the middle increment method, which exhibits such advantages as faster convergence speed and higher efficiency, is applied to stress integration. It is emphasized that compressive stress is positive in the paper.

According to Fig 18, the main computation stages at each site of interest are as follows:

1. The sand content and model parameter are calculated.

2. The tangent modulus ($E_t$) and tangent Poisson's ratio ($v_t$) are computed to obtain the initial stiffness matrix ($D(\varepsilon_0, \sigma_0)$) based on the initial stress state of the incremental step.

3. Based on the initial strain increment ($\Delta\varepsilon$), the first stress increment ($\Delta\sigma_1 = D(\varepsilon_0, \sigma_0)\Delta\varepsilon$) is calculated.

4. The average stress and strain ($\bar{\sigma} = \sigma_0 + 0.5\Delta\sigma_1, \bar{\varepsilon} = \varepsilon_0 + 0.5\Delta\varepsilon_1$) in the incremental step are determined to obtain the new stiffness matrix ($D(\bar{\varepsilon}, \bar{\sigma})$).

5. The formal stress increment ($\Delta\sigma = D(\bar{\varepsilon}, \bar{\sigma})\Delta\varepsilon$) is estimated on the basis of the initial strain increment ($\Delta\varepsilon$).

6. The stress increment and strain increment ($\sigma = \sigma_0 + \Delta\sigma, \varepsilon = \varepsilon_0 + \Delta\varepsilon$) are updated and assigned to the Jacobi matrix DDSDDE.

### Model verification

To evaluate the performance of the proposed sand-content-dependent constitutive model, A series of separate CD tests were conducted on sand-fine mixtures with different sand contents under the confining pressure of 500 kPa. Utilizing ABAQUS software, a three-dimensional finite element model of the sample was created, as displayed in Fig 19. The model, which had boundary conditions similar to those of the triaxial test and had a diameter of 50mm and a

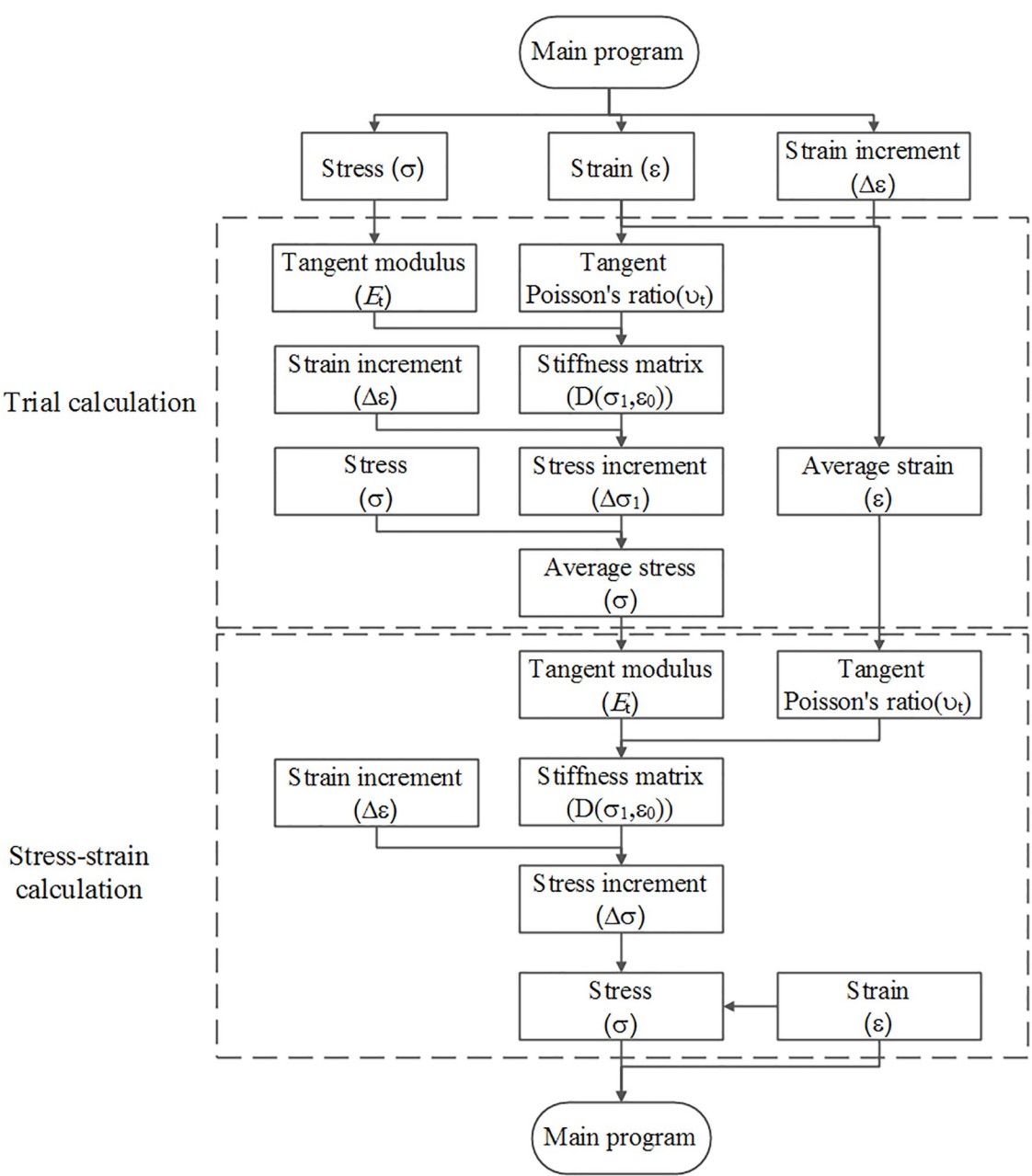

**Fig 18. Flow chart of UMAT development.**

height of 100 mm, was depicted in Fig 19. The calculation element was selected as three-dimensional stress 8-node element (C3D8) made out of a hexagonal mesh with 19885 nodes and 18080 components. Table 8 presents the related model parameters. Fig 20 shows a comparison of the stress-strain curves obtained from experimental measurements and calculations.

As can be observed from Fig 20 (S20 Table in S1 File), measurements have good consistency with calculations, revealing that the sand-content-dependent stress-strain model can properly predict how the changes in sand content would affect the machine behavior of sand-silt

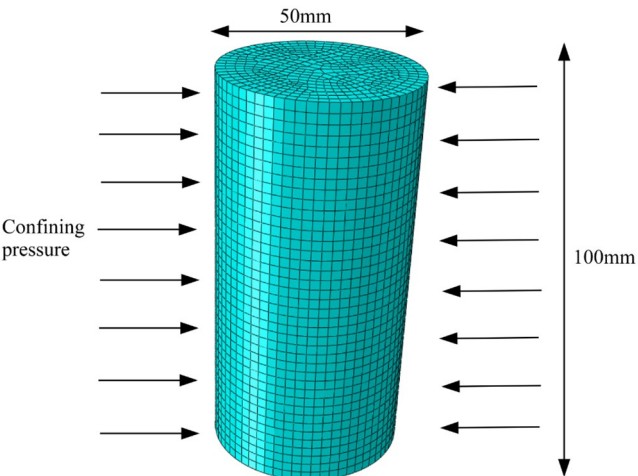

**Fig 19. Schematic diagram of the model.**

mixtures. The errors between the two analyses for axial strain versus volumetric strain for sand-fine mixtures with a sand content of 60% are relatively large when the axial strain is>20%.

## Discussion and conclusion

### Discussion

Numerous studies [23–25] have revealed that there are specific sand content levels in sand-fine mixtures at which the mixtures' shear strength is either fully or partially controlled by the sand content or is completely uncontrolled. In order to explain the phenomenon, Vallejo [23, 24] and Thevanayagam [26] developed a conceptual framework (shown in Fig 21) in which it is assumed that the soil mixture is made up of spherical particles with two different diameter values, sand grains and fine grains.

In Fig 21, the relevant roles of sand and fine grains in the aforementioned conceptual framework's four limiting cases of microstructure are stated. In case (i), the sand content ($F_s$) exceeds a certain threshold value, and the fine grains are completely enclosed within the void spaces between the sand grains. In this case, the sand grain contacts are principally in charge of controlling the mechanical behavior. The inter-sand void ratio ($e_s$) can therefore be used as an index of active particle interactions while ignoring the impacts of fines. While the global void has grown, the sand content in cases (ii) and (iii) is still higher than the threshold value. In these cases, the inter-sand grain interactions still have a big impact. However, while the

**Table 8. Calculated model parameters for sand-fine mixtures under the confining pressure of 500 kPa.**

| Sand content | $E_s$ | $B_s$ | $\varepsilon_{af}$ | $M_{1s}$ | $M_{2s}$ | $M_{3s}$ |
|---|---|---|---|---|---|---|
| /% | | | /% | $/10^{-4}$ | | $/10^{-2}$ |
| 0 | 793.021 | 1048.767 | 1.51 | -4.425 | -0.128 | 1.837 |
| 16.67 | 312.110 | 842.460 | 1.57 | -5.832 | -0.152 | 2.434 |
| 18.57 | 243.131 | 886.525 | 1.73 | -7.251 | -0.191 | 3.086 |
| 50 | 1123.595 | 1030.928 | 1.42 | 11.070 | -0.342 | 4.732 |
| 60 | 678.426 | 856.898 | 1.57 | 13.600 | -0.470 | 5.396 |

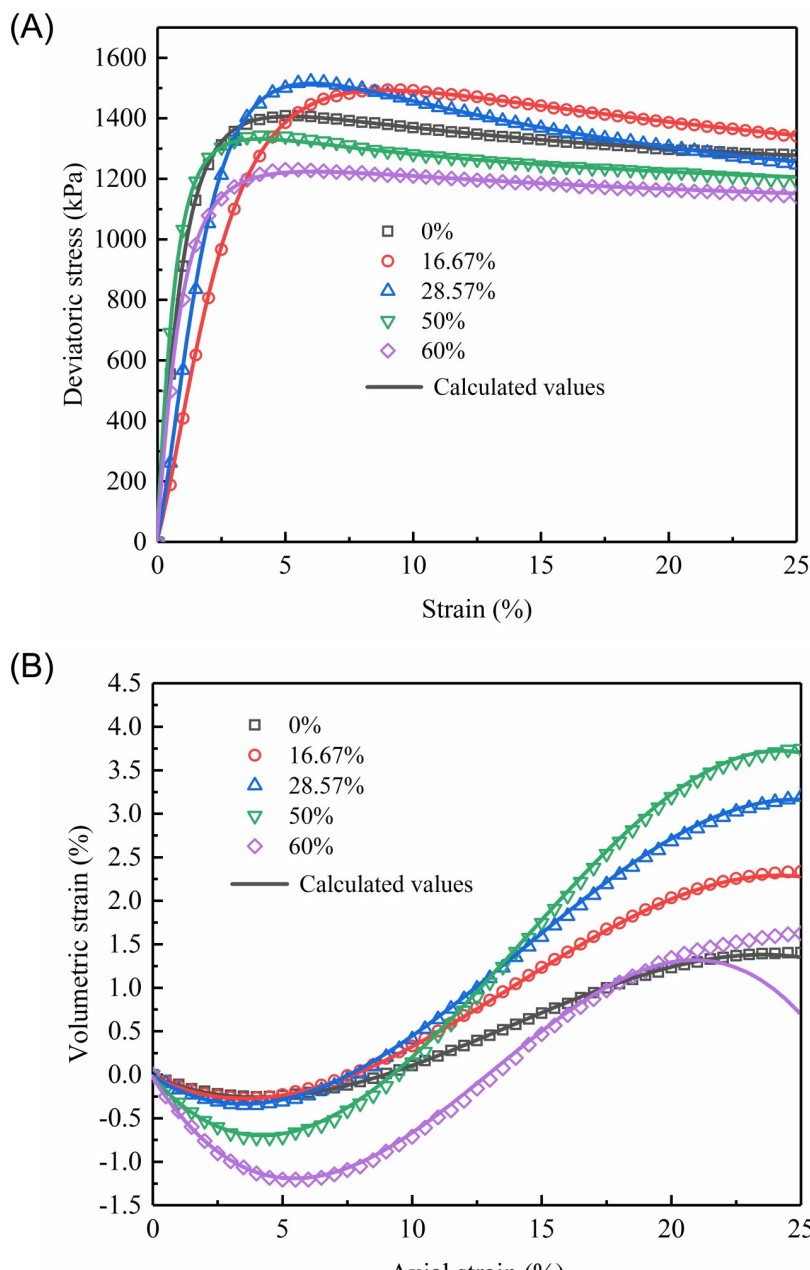

**Fig 20. Comparison between experimental measurements and calculations for sand-silt mixtures with different sand contents under the confining pressure of 500 kPa: (A) Axial strain versus deviatoric stress; (B) Axial strain versus volumetric strain.**

remaining fines (referred to as confined fines) fill the voids between the sand grains, some of the fine grains (referred to as separating fines) become active participants in the internal force chain. Finally, In case (iv), the amount of sand content is less than the threshold value, and the fine grains start to dominate while the sand grains start to play a less role. Neglecting the effects of dispersed sand grains, the inter-fine void ratio ($e_f$) can be utilized as an indicator of active particle interactions.

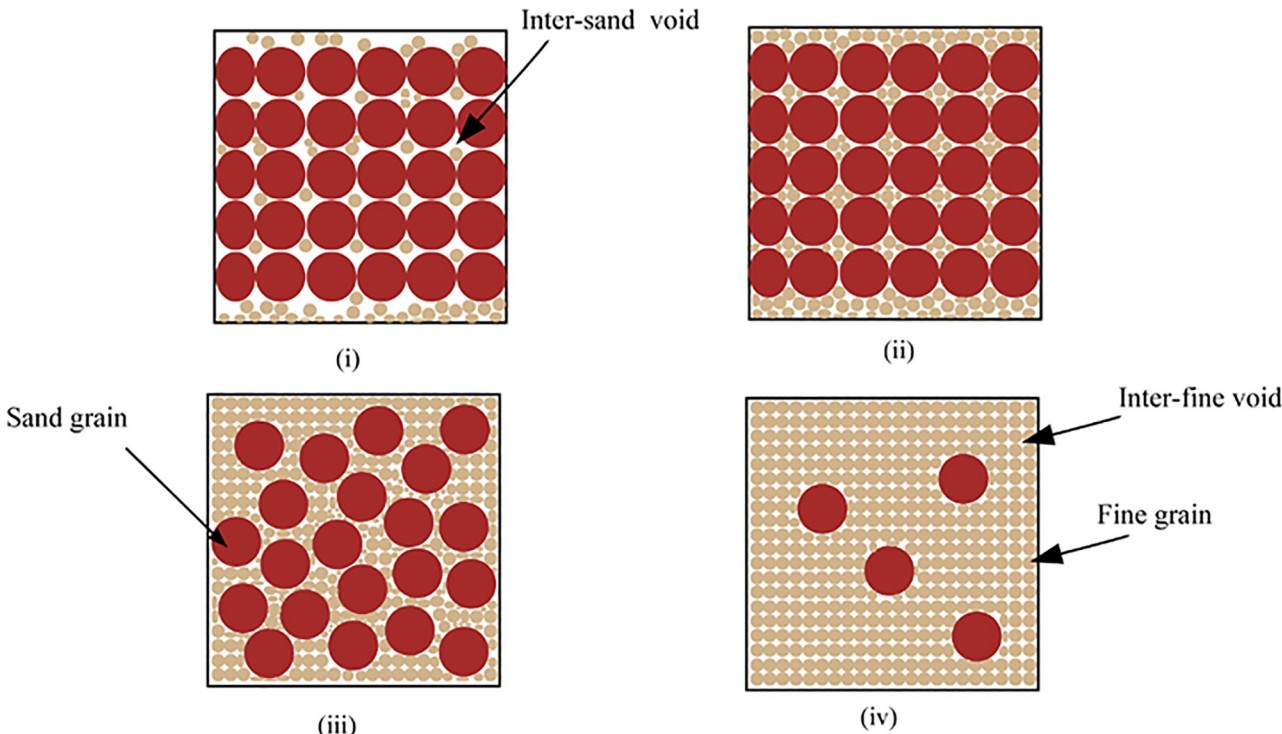

**Fig 21. Schematic diagram of the microstructure of sand-fine mixtures for the conceptual framework [26].**

Based on the aforementioned conceptual framework, it can be consequently shown that the presence of sand grains may increase or decrease the machine behavior of sand-fine mixtures depending on both $e_s$ and $e_f$. Hence, the inter-sand void ratio ($e_s$) and inter-fine void ratio ($e_f$), regarded as the indexes of active particle contacts, will be introduced into the modified Duncan-Chang model, which is used in the quantitative description for the import of sand content on the mechanical properties of sand-fine mixtures in the next work.

It is clear from the illustrations in Figs 10, 11, and 20 that the modified Duncan-Chang model can reproduce the strain-softening characteristics of coral clay, undisturbed loess, and sand-fine mixtures. However, the stress-strain relations of geomaterials have softening type and hardening type [14]. In order to describe these different types of curves in a unified way, the reliability of the modified Duncan-Chang model in describing the hardening stress-strain curves will also be discussed in the next work.

## Conclusion

In this paper, a series of CD tests were conducted on reconstructed sand-fine mixtures to the effect of the sand content on the machine characteristics of the soil. A modified Duncan-Chang model was proposed to characterize the strain-softening behavior of geomaterials based on the initial Duncan-Chang model. A sand-content-dependent constitutive model based on the modified Duncan-Chang model was then developed, which considered the effects of the confining pressure and sand content of the soil by constructing the relationship between model parameters and confining pressure and sand content. The model was implemented as a user-defined model in the ABAQUS software and validated using the simulation of a series of CD tests. The conclusions are as follows:

1. Under different confined pressures, the stress-strain curves of sand-fine mixtures with different sand contents tend to soften, and the softening degree gradually increases with the increase of confining pressure. The soil undergoes shear dilation under the axial load.

2. Compared with the initial Duncan-Chang model, the stress-strain curves described by the modified Duncan-Chang model are very close to the measured stress-strain curves, indicating that the modified model can represent precisely the strain-softening characteristics of coral clay and undisturbed loess.

3. The sand-content-dependent constitutive model is used to simulate the triaxial test data of sand-fine mixtures with different sand contents under the confining pressure of 500 kPa. Both computer simulation and actual measurement indicate that the constitutive model can quantitatively describe the effect of sand content on the mechanical behavior of sand-fine mixtures.

## Supporting information

**S1 File.**
(RAR)

## Acknowledgments

The authors would like to thank the Tongxin County Water Affairs Bureau for approving the field site access and for their help with the field sampling.

## Author Contributions

**Data curation:** Liucheng Chang, Ya Wang.

**Funding acquisition:** Hongyu Wang.

**Investigation:** Liucheng Chang, Hongyu Wang.

**Methodology:** Liucheng Chang.

**Project administration:** Hongyu Wang.

**Software:** Liucheng Chang.

**Validation:** Liucheng Chang.

**Writing – original draft:** Liucheng Chang.

**Writing – review & editing:** Ya Wang, Hongyu Wang.

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
