## [Decision Letter · Decision Letter 0]

3 Apr 2023

PONE-D-23-02538Influence of sand content on the mechanical properties of sand-silt mixtures from check dam deposits in the Loess Hilly of Ningxia,ChinaPLOS ONE

Dear Dr. Wang,

Thank you for submitting your manuscript to PLOS ONE. After careful consideration, we feel that it has merit but does not fully meet PLOS ONE’s publication criteria as it currently stands. Therefore, we invite you to submit a revised version of the manuscript that addresses the points raised during the review process.

ACADEMIC EDITOR: You'll notice that the reviewers are advising a "major revision" of your paper before it can be processed further. Therefore, be sure to fully address all the issues that the reviewers, particularly reviewer #2, raised.

Note: If a reviewer requests that you reference their articles or the articles of their affiliates, you are not required to comply. Consider only works of writing that are pertinent to your own research.

We look forward to receiving your revised manuscript.

Kind regards,

Paul Awoyera

Academic Editor

PLOS ONE

“The study was supported by the Natural Science Foundation of China (Project No.41962016) and the Doctoral Scientific Fund Project of the Ministry of Education of China (Project No.20136401110003)”

5. Please include a copy of Table 4 which you refer to in your text on page 12.

Reviewers' comments:

Reviewer's Responses to Questions

**Comments to the Author**

1. Is the manuscript technically sound, and do the data support the conclusions?

Reviewer #1: Yes

Reviewer #2: No

2. Has the statistical analysis been performed appropriately and rigorously? 

Reviewer #1: Yes

Reviewer #2: No

3. Have the authors made all data underlying the findings in their manuscript fully available?

Reviewer #1: Yes

Reviewer #2: Yes

4. Is the manuscript presented in an intelligible fashion and written in standard English?

Reviewer #1: Yes

Reviewer #2: Yes

5. Review Comments to the Author

Reviewer #1: I consider this work novel. In general, the paper is well organized and written. The following comments and suggestions are provided for the author's revision:

Comments:

1. Abstract is not written appropriately. More precise results should be summarized

2. Novelty and significance of this study should be addressed.

3. Comparing fitted results to the measured ones is not adequate.

4. The literature review lacks thorough understanding on the recent developments in this research field. ground improvement methods should be explained

A. Vafaei, A., Choobbasti, A. J., Koutenaei, R. Y., Vafaei, A., Afrakoti, M. P., & Kutanaei, S. S. (2023). Effect of Barley Straw Fiber as a Reinforcement on the Mechanical Behavior of Babolsar Sand. Transportation Infrastructure Geotechnology, 1-20.

B. Ghadakpour, M., Fakhrabadi, A., Janalizadeh Choobbasti, A., Soleimani Kutanaei, S., Vafaei, A., Taslimi Paein Afrakoti, M., & Eisazadeh, N. (2022). Effect of post-construction moisture condition on mechanical behaviour of Fiber-reinforced-cemented-sand (FRCS). Geomechanics and Geoengineering, 17(6), 1852-1864.

C. Roshan, K., Choobbasti, A. J., Kutanaei, S. S., & Fakhrabadi, A. (2022). The effect of adding polypropylene fibers on the freeze-thaw cycle durability of lignosulfonate stabilised clayey sand. Cold Regions Science and Technology, 193, 103418.

D. Ghadakpour, M., Fakhrabadi, A., Janalizadeh Choobbasti, A., Soleimani Kutanaei, S., Vafaei, A., Taslimi Paein Afrakoti, M., & Eisazadeh, N. (2021). Effect of post-construction moisture condition on mechanical behaviour of Fiber-reinforced-cemented-sand (FRCS). Geomechanics and Geoengineering, 1-13.

E. Roshan, K., Choobbasti, A. J., & Kutanaei, S. S. (2020). Evaluation of the impact of fiber reinforcement on the durability of lignosulfonate stabilized clayey sand under wet-dry condition. Transportation Geotechnics, 23, 100359.

F. Afrakoti, M. T. P., Choobbasti, A. J., Ghadakpour, M., & Kutanaei, S. S. (2020). Investigation of the effect of the coal wastes on the mechanical properties of the cement-treated sandy soil. Construction and Building Materials, 239, 117848.

G. Vafaei, A., Choobbasti, A. J., Koutenaei, R. Y., Vafaei, A., Afrakoti, M. P., & Kutanaei, S. S. (2022). Experimental investigation of the mechanical behavior and engineering properties of sand reinforced with hemp fiber. Arabian Journal of Geosciences, 15(22), 1679.

5. The paper reads like a short report. The authors must improve the discussion in the paper. Avoid general conclusions and mention limitations of their work.

6. Validity and repeatability of the experiments and results should be checked.

7. A detailed experimental procedure and the accuracy of different components of the experimental system are given. Measurement equipments should be described. And also, the characteristics of measurement equipments should be given.

Reviewer #2: The paper studied the influence of sand content on the mechanical properties of sand-silt mixtures from check dam deposits in the Loess Hilly of Ningxia,China, in which the CD tests were performed and Duncan-Chang model was revised to describe the stress-strain relationship. The volumetric strain is not provided there, and the samples behave strain softening, which can not described the Duncan-chang model. In view of this, the paper can not be accepted for publication.

6. PLOS authors have the option to publish the peer review history of their article (what does this mean?). If published, this will include your full peer review and any attached files.

Reviewer #1: No

Reviewer #2: No

---

## [Author Response · Author response to Decision Letter 0]

30 May 2023

Reviewer #1: 

1. Abstract is not written appropriately. More precise results should be summarized.

Response: Considering the reviewer’s suggestion, we have rewritten the abstract, and the important conclusions of the paper are summarized in the Abstract.

2. Novelty and significance of this study should be addressed. 

Response: Considering the reviewer’s suggestion, the novelty and significance of the study have been addressed in the Research Significance.

3. Comparing fitted results to the measured ones is not adequate. 

Response: Considering the reviewer’s suggestion, Adding two new cases is to verify the reliability of the Modified Duncan-Chang model proposed in this paper.

4. The literature review lacks thorough understanding on the recent developments in this research field. ground improvement methods should be explained. 

Response: Thank you for the references provided by the reviewers, some research about the engineering characteristics of sand-fine mixtures in the past three years has been added to the literature review. 

5. The paper reads like a short report. The authors must improve the discussion in the paper. Avoid general conclusions and mention limitations of their work. 

Response: Considering the reviewer’s suggestion, we have supplemented the discussion section in the paper and the limitations of our work are mentioned in the Discussion. The main conclusions of the paper have been summarized in the Conslusion. 

6. Validity and repeatability of the experiments and results should be checked.

Response: Considering the reviewer’s suggestion, the validity and repeatability of the experiments and results have been checked.

7. A detailed experimental procedure and the accuracy of different components of the experimental system are given. Measurement equipments should be described. And also, the characteristics of measurement equipments should be given.

Response: Considering the reviewer’s suggestion, the detailed experimental procedure and the accuracy of different components of the experimental system have been described in the Experimental Equipment and Experimental Procedures, respectively. The characteristics of measurement equipment have been also described in the Experimental Equipment. 

Special thanks to you for your good comments.

Reviewer #2: 

1. The paper studied the influence of sand content on the mechanical properties of sand-silt mixtures from check dam deposits in the Loess Hilly of Ningxia, China, in which the CD tests were performed and the Duncan-Chang model was revised to describe the stress-strain relationship. The volumetric strain is not provided there, and the samples behave strain softening, which can not describe the Duncan-chang model. In view of this, the paper can not be accepted for publication.

Response: It is really true as Reviewer suggested that the volumetric strain is not provided in the paper. Considering the reviewer’s suggestion, we have supplemented this part of the experimental data in the paper. As the reviewer said the initial Duncan-Chang model can not characterize the softening stress-strain curves of geomaterials. However, the results of the triaxial test in the study show that the stress-strain relationships of sand-fine mixtures with different sand contents under various confining pressures have the characteristic of strain softening. In order to apply the Duncan-Chang model to the quantitative analysis of the effect of sand content on the mechanical properties of sand-fine mixtures, the model must be further modified to reflect the strain softening behavior of the soil. Then a modified Duncan-Chang model is developed based on the initial Duncan-Chang model and its reliability in describing the softening stress-strain curves is verified by comparing the predicted stress-strain curves with the measured curves of coral clay and undisturbed loess. Finally, a sand-content-dependent constitutive model that considered the effects of sand content and confining pressure of the soil by constructing the relationship between model parameters and confining pressure and sand content is proposed based on the modified Duncan-Chang model. The constitutive model is implemented in ABAQUS (version 6.14) software and verified by comparing the calculated results with the triaxial test data of sand-fine mixtures under the confining pressure of 500 kPa. The comparison results indicate that the constitutive model can reflect the real characteristics of sand-fine mixtures.

Special thanks to you for your good comments.

1. Thank you for stating the following financial disclosure:

"The study was supported by the Natural Science Foundation of China (Project No.41962016) and the Doctoral Scientific Fund Project of the Ministry of Education of China (Project No.20136401110003)"

Response: The Natural Science Foundation of China (Project No.41962016) and the Doctoral Scientific Fund Project of the Ministry of Education of China (Project No.20136401110003) play an important role in the study design, data collection and analysis, and preparation of the manuscript, and the amended role of these funders have been stated in the cover letter. 

2. Please include a copy of Table 4 which you refer to in your text on page 12.

Response: The copy of Table 4 on page 12 is within the S1 files.

Response: There are no permits which are required for the word. This is because all work is completed by myself and other research group members without involving third-party organizations.

---

## [Decision Letter · Decision Letter 1]

7 Jun 2023

Influence of sand content on the mechanical properties of sand-silt mixtures from check dam deposits in the Loess Hilly of Ningxia,China

PONE-D-23-02538R1

Dear Dr. Wang,

We’re pleased to inform you that your manuscript has been judged scientifically suitable for publication and will be formally accepted for publication once it meets all outstanding technical requirements.

Kind regards,

Paul Awoyera

Academic Editor

PLOS ONE

Additional Editor Comments (optional):

Reviewers' comments:

Reviewer's Responses to Questions

**Comments to the Author**

1. If the authors have adequately addressed your comments raised in a previous round of review and you feel that this manuscript is now acceptable for publication, you may indicate that here to bypass the “Comments to the Author” section, enter your conflict of interest statement in the “Confidential to Editor” section, and submit your "Accept" recommendation.

Reviewer #1: All comments have been addressed

Reviewer #2: All comments have been addressed

2. Is the manuscript technically sound, and do the data support the conclusions?

Reviewer #1: Yes

Reviewer #2: Yes

3. Has the statistical analysis been performed appropriately and rigorously? 

Reviewer #1: Yes

Reviewer #2: Yes

4. Have the authors made all data underlying the findings in their manuscript fully available?

Reviewer #1: Yes

Reviewer #2: Yes

5. Is the manuscript presented in an intelligible fashion and written in standard English?

Reviewer #1: Yes

Reviewer #2: Yes

6. Review Comments to the Author

Reviewer #1: All comments have been addressed. The article has the necessary quality to be accepted in this journal

Reviewer #2: The authors have revised the paper according to the comments and all the commnets have been addressed. It can be accepted for publication.

7. PLOS authors have the option to publish the peer review history of their article (what does this mean?). If published, this will include your full peer review and any attached files.

Reviewer #1: No

Reviewer #2: No

---

## [Editor Report · Acceptance letter]

23 Oct 2023

PONE-D-23-02538R1 

Influence of sand content on the mechanical properties of sand-silt mixtures from check dam deposits in the Loess Hilly of Ningxia, China 

Dear Dr. Wang:

I'm pleased to inform you that your manuscript has been deemed suitable for publication in PLOS ONE. Congratulations! Your manuscript is now with our production department. 

Kind regards, 

on behalf of

Dr. Paul Awoyera 

Academic Editor

PLOS ONE